# Genomic adaptation to small population size and saltwater consumption in the critically endangered Cat Ba langur

Liye Zhang [1,2,3] ✉, Neahga Leonard [4], Rick Passaro[4], Mai Sy Luan[4], Pham Van Tuyen[4], Le Thi Ngoc Han[4], Nguyen Huy Cam[4], Larry Vogelnest[5], Michael Lynch[6], Amanda E. Fine [7], Nguyen Thi Thanh Nga[8], Nguyen Van Long[8], Benjamin M. Rawson [9], Alison Behie[10], Truong Van Nguyen [1,11,12], Minh D. Le[12,13], Tilo Nadler[14], Lutz Walter[1], Tomas Marques-Bonet [15,16,17,18], Michael Hofreiter [11] ✉, Ming Li [3] ✉, Zhijin Liu[19] ✉ & Christian Roos [1,20] ✉

Many mammal species have declining populations, but the consequences of small population size on the genomic makeup of species remain largely unknown. We investigated the evolutionary history, genetic load and adaptive potential of the Cat Ba langur (*Trachypithecus poliocephalus*), a primate species endemic to Vietnam's famous Ha Long Bay and with less than 100 living individuals one of the most threatened primates in the world. Using high-coverage whole genome data of four wild individuals, we revealed the Cat Ba langur as sister species to its conspecifics of the northern limestone langur clade and found no evidence for extensive secondary gene flow after their initial separation. Compared to other primates and mammals, the Cat Ba langur showed low levels of genetic diversity, long runs of homozygosity, high levels of inbreeding and an excess of deleterious mutations in homozygous state. On the other hand, genetic diversity has been maintained in protein-coding genes and on the gene-rich human chromosome 19 ortholog, suggesting that the Cat Ba langur retained most of its adaptive potential. The Cat Ba langur also exhibits several unique non-synonymous variants that are related to calcium and sodium metabolism, which may have improved adaptation to high calcium intake and saltwater consumption.

As a result of human population expansion, habitat transformation, and other human activities including direct persecution, many species are on the brink of extinction or have declining populations[1–6]. Non-human primates are no exception as 63% of the species are threatened with extinction and 93% have declining populations[7,8]. Threats to primates include, among others, destruction and fragmentation of habitat, hunting, and poaching[7].

Small populations are vulnerable to a range of extrinsic and intrinsic processes, including habitat loss and disease, demographic stochasticity as well as genetic effects[9]. In recent years, an increasing number of studies focused on the role of genetics on the long-term viability of small populations and often revealed genomic erosion in such populations[10–21]. Small populations tend to have reduced viability due to a loss of genetic diversity and increased inbreeding[22–25]. Moreover, in small, isolated, and inbred populations, genetic load, the presence of unfavorable genetic material, may accumulate and increase extinction risk, although purging of deleterious mutations has been documented, too[12–21,26–28].

A full list of affiliations appears at the end of the paper. ✉ e-mail: lzhang@dpz.eu; michael.hofreiter@uni-potsdam.de; lim@ioz.ac.cn; liuzj6888@cnu.edu.cn; croos@dpz.eu

The critically endangered golden-headed or Cat Ba langur (*Trachypithecus poliocephalus*), endemic to Cat Ba Island in Ha Long Bay, northeastern Vietnam (Fig. 1a, b) and one of the most threatened primates in the world[29], is a good model to study the effects of small population size and recent population declines on the genomic makeup of a species. The population may have contained 2400–2700 individuals in historical times[30]. However, the first survey conducted in 1999 revealed only 104–135 individuals[30], and in 2002–2004, the population declined to a minimum of 40 individuals[29]. Since then, the population increased to 74–79 individuals of which 38 are reproductively active (status: December 2023; NL pers. observation). Major reasons for the decline both in the past and until recently were hunting and poaching for traditional medicine and sport, while today the species suffers mainly from disturbance and fragmentation of habitat, growing, but poorly managed tourism, and inbreeding[29].

*Trachypithecus poliocephalus* is a species of the colobine genus *Trachypithecus* which contains a total of 22 species, grouped into four species groups[31–34]. *Trachypithecus poliocephalus* is one of the seven species of the *T. francoisi* or limestone langur group[31–34]. Within this group, *T. poliocephalus* forms the northern clade, together with François's langur (*T. francoisi*) and the white-headed langur (*T. leucocephalus*), but it remains unclear whether the species is basal within this clade as suggested by mitochondrial DNA[34] or it is sister to *T. leucocephalus* as indicated by similarities in fur coloration, which led Groves[35] to treat them both as subspecies of a single species. As all species of the limestone langur group, *T. poliocephalus* is restricted to limestone karst habitats – in contrast to other species of *Trachypithecus* that live in rainforest habitats[31–34]. Although it has been suggested that limestone langurs originally may have also occurred in rainforests and survived only in karst habitats due to human pressure[36], a recent genomic study revealed evidence for genomic adaptation of this group to karst habitat, specifically to high calcium intake, since more than one million years[37]. However, even among limestone langurs and also primates in general, *T. poliocephalus* is probably unique as it seems to be able to cope with high salt concentrations in its diet. As the only limestone langur species living on a

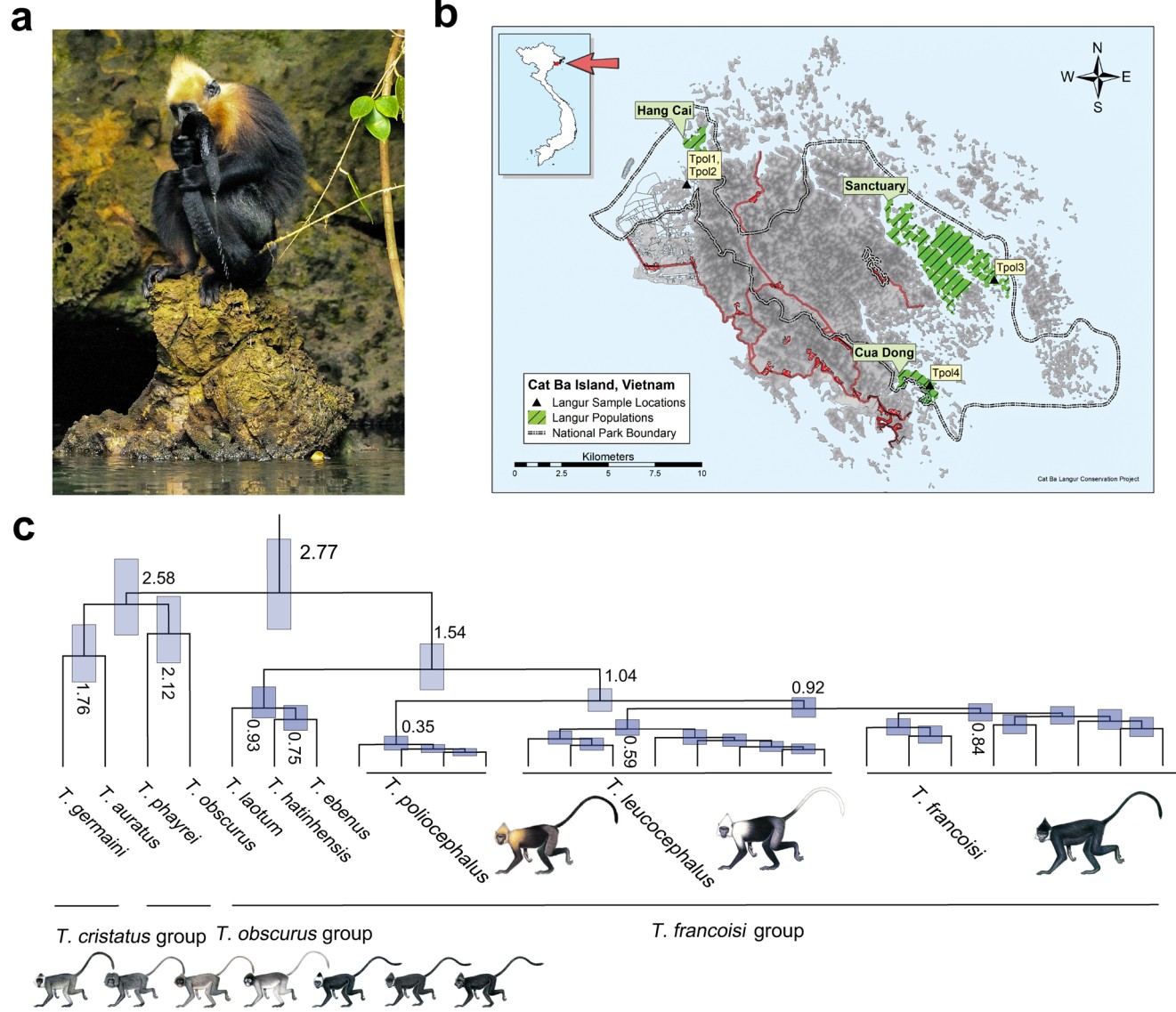

**Fig. 1 | Photo of a Cat Ba langur, sampling sites on Cat Ba Island, and dated phylogeny of *Trachypithecus* langurs. a** *Trachypithecus poliocephalus* individual licking saltwater from its tail (Photo: Nguyen Van Truong). **b** Sampling sites of the four *T. poliocephalus* individuals (Tpol1-4) on Cat Ba Island, Vietnam. **c** Ultrametric tree showing phylogenetic relationships and divergence times among *Trachypithecus* langurs. All nodes are supported by bootstrap values of >95%. Numbers at nodes refer to million years ago and the blue bars indicate 95% confidence intervals (for details see Supplementary Fig. 3; Supplementary Table 4). Primate illustrations copyright 2024 Stephen D. Nash (IUCN SSC Primate Specialist Group). Used with permission.

maritime island, *T. poliocephalus* is naturally exposed to high salt concentrations in the form of saline-rich drinking water and moisture on food plants, and animals are known to lick and even drink sea water[38] (Fig. 1a; Supplementary Movie 1). However, so far evidence for genomic adaptation to an increased saltwater tolerance in *T. poliocephalus* is missing.

In this work, we examine the effects of small population size on the conservation status of *T. poliocephalus* by analyzing whole-genome data of four wild individuals. We first investigate the phylogenetic relationships among northern limestone langur species, estimate when they diverged, and if secondary gene flow occurred. Second, as a proxy for adaptive potential, we calculate the genomic diversity and inbreeding level for *T. poliocephalus* and compare them with data of other limestone langurs and further mammal species. Third, we examine the genetic load of *T. poliocephalus* in comparison to its conspecifics of the northern limestone langur clade. Finally, we investigate signatures of selection, which are potentially associated with adaptation to the species' unique environment. Our results provide insights into the evolutionary history of limestone langurs, how small population size affects the genomics of a critically endangered primate, and how species adapt to challenging environmental conditions.

## Results

### Sampling and datasets

We generated whole-genome sequencing data (range: 28.8–36.5×, mean: 32.7×; Supplementary Table 1) from four wild *T. poliocephalus* individuals representing all three extant sub-populations of the species (Hang Cai, Sanctuary, Cua Dong; Fig. 1b). Due to the expected high inbreeding level in the population[29], we tested for relatedness among the four *T. poliocephalus* individuals, but found no evidence for any close relatedness among them (Supplementary Table 2). For comparative analyses, we added additional published genome data and mapped all samples to the *T. francoisi* (Tfra_2.0) and rhesus macaque (*Macaca mulatta*; Mmul_10) reference genomes (see "Methods" section). To avoid any bias arising from mapping to an ingroup and to take advantage of the high-quality chromosome-level annotation of Mmul_10, we performed most analyses using the mapping data to Mmul_10 (for details see Supplementary Table 3). However, for genome-wide analyses of length and fraction of runs of homozygosity (ROHs), and signatures of selection, we used the mapping data to the more closely related Tfra_2.0 reference genome.

### Phylogeny and population structure

We explored the phylogenetic position of *T. poliocephalus* by reconstructing neighbor-joining (NJ) and maximum-likelihood (ML) trees based on autosomal nucleotide variants. Both trees supported *T. poliocephalus* as the sister lineage to *T. francoisi* and *T. leucocephalus* (NJ and ML bootstrap values of 100% and >95%, respectively; Fig. 1c; Supplementary Figs. 1 and 2). Together, these three species formed the northern clade of limestone langurs and represented the sister group to the southern clade consisting of *T. laotum*, *T. hatinhensis* and *T. ebenus*. Using protein-coding sequences (CDS) on autosomes, we estimated the initial split among the investigated *Trachypithecus* species at 2.8 (95% confidence interval: 2.2–3.5) million years ago (Mya; Fig. 1c; Supplementary Fig. 3, Supplementary Table 4), separating limestone langurs from two other species groups of the genus (*T. obscurus* and *T. cristatus* groups). Among limestone langurs, southern and northern clades diverged 1.5 (1.2–2.0) Mya, and in the northern clade, *T. poliocephalus* separated from *T. francoisi* and *T. leucocephalus* 1.0 (0.8–1.3) Mya, while the latter two diverged 0.9 (0.8–1.1) Mya. Admixture plots and a principal component analysis (PCA) supported the division of the three northern limestone langur species into three clusters and the basal position of *T. poliocephalus* among them (Supplementary Figs. 4 and 5; Supplementary Table 5). We further tested for gene flow events among these three species, but found no evidence

for extensive, post-divergence gene flow among them (*D*-statistics; no significant difference under the $X^2$ test, $p = 0.740 > 0.05$; Supplementary Tables 6 and 7).

### Genetic diversity and inbreeding

As a proxy for their adaptive potential[22], we inferred nucleotide diversity π of limestone langurs and compared it with those of other mammals. Results showed that nucleotide diversity of *T. poliocephalus* was with 0.033% one of the lowest among a set of 55 mammal species (Fig. 2a). Likewise, genome-wide autosomal heterozygosity *He* of *T. poliocephalus* was with 0.36 (0.35–0.38) heterozygous sites per 1000 bp the lowest among limestone langurs (Fig. 2b). However, although in all limestone langurs heterozygosity was generally lower in protein-coding (exons) versus non-protein-coding regions, *T. poliocephalus* exhibited in exons the highest heterozygosity rate among limestone langurs (One-way ANOVA test, $p < 0.001$; Supplementary Fig. 6; Supplementary Data 1). Also, the ratio of non-synonymous to synonymous variants was increased in *T. poliocephalus* compared to the other two species, although not significant compared to *T. francoisi* ($p < 0.001$ vs. *T. leucocephalus*; Supplementary Fig. 7; Supplementary Data 1).

Next, we investigated ROHs, contiguous homozygous segments of the genome where identical haplotypes are inherited from both parents and which give insights into the degree of inbreeding[39,40]. The longest ROHs were identified in *T. poliocephalus* (25.48–40.16 Mb), followed by *T. leucocephalus* (10.31–22.81 Mb), *T. francoisi* (7.68–16.59 Mb), *T. laotum* (13.11 Mb), *T. hatinhensis* (11.12 Mb), and *T. ebenus* (7.09 Mb). Overall, the *T. poliocephalus* genome contained the lowest number of ROHs, but due to the comparatively large number of long ROHs (>1 Mb), the fraction of the genome in ROHs was with 88.98% the highest among limestone langurs (Fig. 2c; Supplementary Figs. 8–10). Likewise, also the genomic inbreeding coefficient *F*, based on all ROHs, was highest for *T. poliocephalus* ($F_{ROH} = 0.85$; Fig. 2d).

We further inspected nucleotide diversity and ROHs for individual chromosomes. We observed a significantly higher nucleotide diversity (π = 0.075%; One-way ANOVA test, $p < 0.001$) and a significantly smaller fraction of ROHs ($f_{ROH} = 0.42$; $p < 0.001$) for the human chromosome 19 ortholog compared to all other autosomes (Fig. 2e; Supplementary Fig. 11). Likewise, on the human chromosome 19 ortholog the fraction of genes in ROHs was with 0.64 significantly smaller ($p < 0.001$) than on any other autosome (range: 0.74–0.82; Supplementary Table 8).

### Deleterious mutations and genetic load

We first estimated the individual masked and realized load using autosomal polymorphism data and genomic evolutionary rate profiling (GERP) scores. *Trachypithecus poliocephalus* with an average of 0.89 and *T. leucocephalus* with an average of 0.94 showed a significantly lower (One-way ANOVA test, $p < 0.05$) masked load than *T. francoisi* (1.01; Supplementary Fig. 12). However, *T. poliocephalus* with an average of 1.85 had a significantly higher ($p < 0.001$) realized load than the others (*T. leucocephalus*: 1.20; *T. francoisi*: 0.94).

We then investigated the impact of deleterious mutations on protein function using four effect categories (modifier, low, moderate, high) which we obtained by snpEFF annotation. In all three northern limestone langur species, the portion of homozygous and heterozygous deleterious mutations with high impact was less than 0.018% of all deleterious mutations and not significantly different among species, but for those with moderate impact, *T. francoisi* showed a significantly lower rate than the other two (One-way ANOVA test, $p < 0.05$); no significant differences among the three species were found in categories low and modifier (Fig. 3a; Supplementary Data 2 and 3). However, when considering only homozygous deleterious mutations with high impact, *T. poliocephalus* exhibited a significantly ($p < 0.05$) higher rate than *T. francoisi* and *T. leucocephalus*

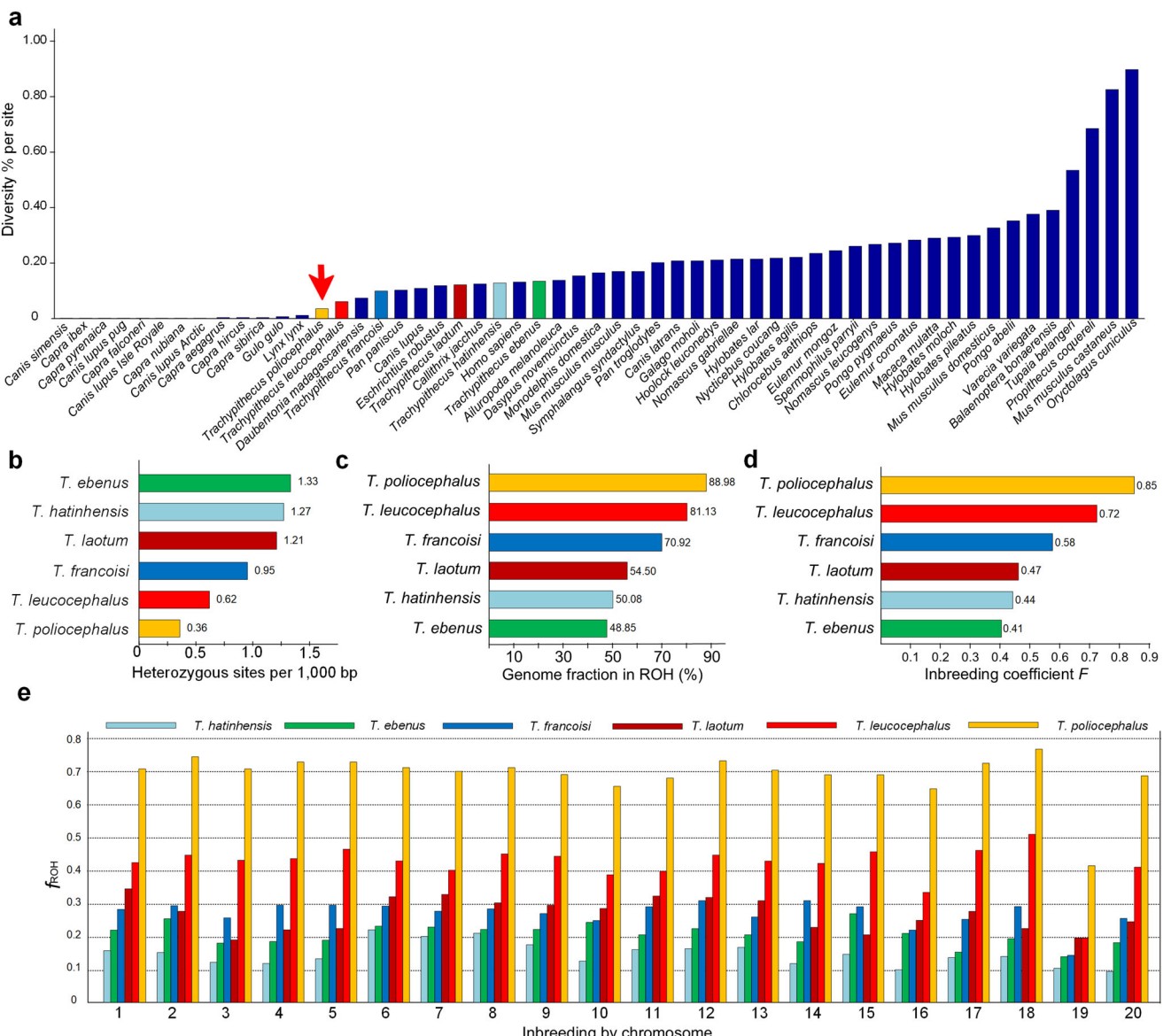

**Fig. 2 | Genetic diversity and inbreeding in Cat Ba langurs. a** Comparison of average nucleotide diversity π between *T. poliocephalus* (orange bar with red arrow) and other mammals (dark blue bars; data from[18,140,141]), including other limestone langurs (differently colored bars; data from this study). **b** Number of heterozygous sites per 1000 bp in limestone langurs. **c** Fraction (%) of the genome in runs of homozygosity (ROHs) in limestone langurs. **d** Inbreeding coefficient *F* for limestone langurs. **e** Fraction of the genome in ROHs $f_{ROH}$ by chromosome in limestone langurs (mapped to Mmul_10). Average chromosome-level $f_{ROH}$ for *T. poliocephalus* (chr1 to 20): 0.72, 0.76, 0.72, 0.74, 0.74, 0.72, 0.71, 0.73, 0.70, 0.67, 0.69, 0.75, 0.72, 0.70, 0.71, 0.66, 0.74, 0.79, 0.42, and 0.70. Source data are provided as a Source Data file.

(Fig. 3b; Supplementary Data 2 and 3). Although not always significant, similar trends with an increased rate of homozygous deleterious mutations in *T. poliocephalus* compared to the other two species were observed also in effect categories moderate and low, while in the category modifier, a lower rate was found (Fig. 3b; Supplementary Data 2 and 3). Functional enrichment analysis revealed that genes containing homozygous high-impact deleterious mutations in *T. poliocephalus* are mainly related to the immune system, signal transduction, RNA metabolism, and gene expression (Supplementary Data 4).

### Positive selection and non-synonymous variants

We identified putative targets of selection between *T. poliocephalus* and *T. francoisi* using strict scan methods (XP-EHH, θπ, Ka/Ks) and revealed a total of 205 candidate genes under strong selection in *T. poliocephalus* compared to *T. francoisi* (Supplementary Fig. 13,

Supplementary Data 5). Functional classification and enrichment analyses showed that many of these genes are related to, among others, metabolism of proteins and RNA, the immune system, organic anion transporters, transcriptional regulation by TP53, and diseases (Benjamini and Hochberg corrected test, $p < 0.05$; Supplementary Data 6).

We further performed whole-genome scans aiming to detect non-synonymous variants that occur only in *T. poliocephalus*. Using this method, we revealed, among others, a total of 92 genes related to calcium pathways 16 genes in the KEGG "calcium signaling pathway" ($p < 0.01$), 71 genes in GO term "calcium ion binding" (corrected $p < 0.001$) and 13 genes in GO term "calcium-mediated signaling" (corrected $p < 0.01$) (Supplementary Table 9, Supplementary Data 7; note that some genes are present in multiple pathways). Seventy of the calcium-related genes (highlighted in red in Supplementary Data 7) contain at least one non-synonymous

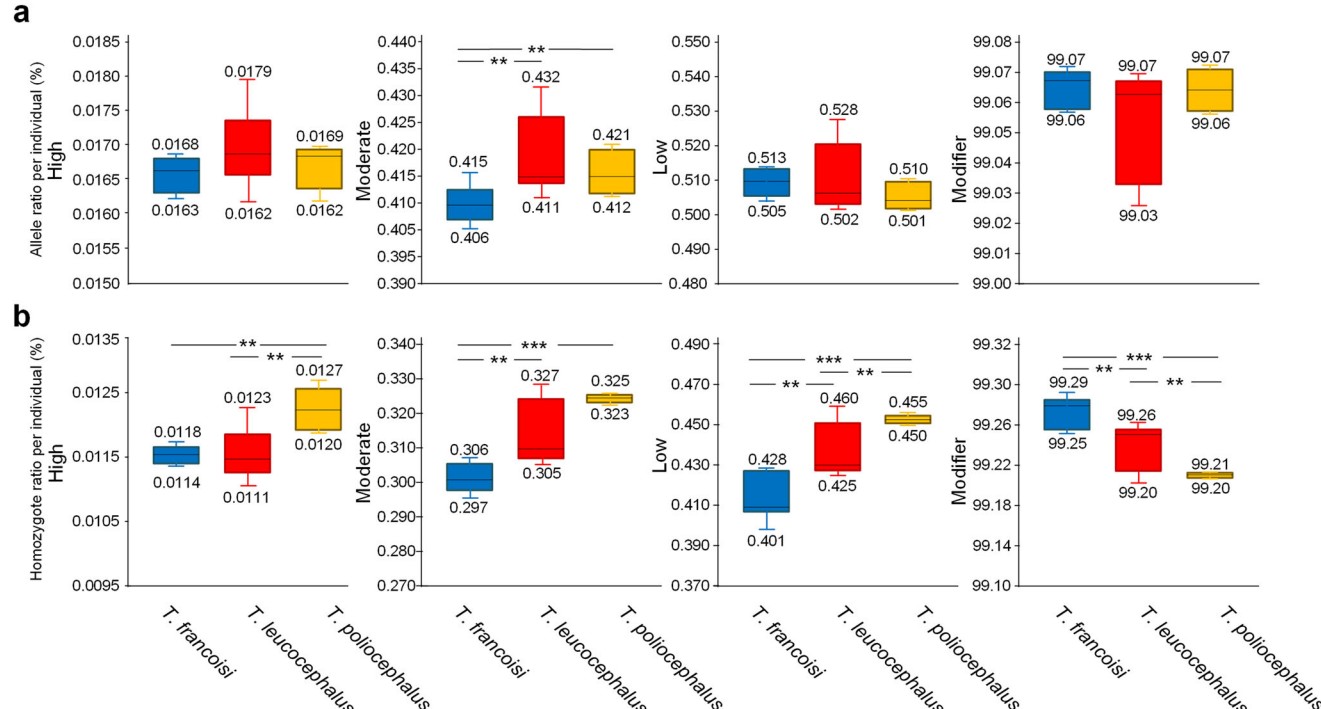

**Fig. 3 | Deleterious mutations in the three northern limestone langurs.** Direct comparison of the effect (high, moderate, low, modifier) of mutations on protein function between *T. francoisi*, *T. leucocephalus* and *T. poliocephalus* based on snpEFF annotation (One-way ANOVA test, n (*T. francoisi*) = 8, n (*T. leucocephalus*) = 8, n (*T. poliocephalus*) = 4, ***p < 0.001, **p < 0.05). **a** Ratio of all alleles (significant p values: moderate impact: $p_{(Tfra-Tleu)}$ = 0.011, $p_{(Tpol-Tfra)}$ = 0.034, $p_{(Tpol-Tleu)}$ = 0.49) and **b** of only homozygotes (significant p values: high impact:

$p_{(Tpol-Tfra)}$ = 0.048, $p_{(Tpol-Tleu)}$ = 0.012; moderate impact: $p_{(Tfra-Tleu)}$ = 0.0022, $p_{(Tpol-Tfra)}$ = 7.01 × 10$^{-7}$; low impact: $p_{(Tfra-Tleu)}$ = 0.0024, $p_{(Tpol-Tfra)}$ = 4.81 × 10$^{-5}$, $p_{(Tpol-Tleu)}$ = 0.045; modifier impact: $p_{(Tfra-Tleu)}$ = 0.0023, $p_{(Tpol-Tleu)}$ = 1.11 × 10$^{-5}$, $p_{(Tpol-Tleu)}$ = 0.043). All p values, minimum value, first quartile (Q1), median (Q2), third quartile (Q3), maximum value, interquartile range (IQR), lower whisker, and upper whisker are provided in the Source Data file.

mutation private to *T. poliocephalus*. In the other 22 genes, we detected amino acid changes that were fixed in *T. poliocephalus*, but occurred with some frequency (max. 28.95%) in other limestone langurs. In 19 (*ACKR3, ACKR4, ADCY1, ASPH, CCKAR, EDN1, EIF2AK3, FPR2, GRIN2D, LAT2, MCTP2, MYLK4, NTSR1, P2RX6, PLA2G4B, PLCD3, SPHK1, TPCN1, TRDN*) of the 24 genes found in GO term "calcium-mediated signaling" and the KEGG "calcium signaling pathway" (some genes are present in both pathways), homozygous non-synonymous mutations were private to *T. poliocephalus* (Supplementary Fig. 14, Supplementary Data 7), while in five genes (*PLCG2, GRIN2C, LAP3, PDE1A, RYR1*) these mutations occurred (max. frequency 10.53%) also in other *Trachypithecus* species (Supplementary Fig. 14, Supplementary Data 7). We further identified 30 non-synonymous mutations in 22 genes related to sodium transport and homeostasis, of which nine genes (*SLC4A4, SLC4A11, SLC38A7, SLC34A3, SLC5A1, SLC5A2, SCNN1D, SLC5A6, SCN5A*) are linked to GO term "sodium ion transport" (corrected p < 0.05; Supplementary Data 8, Supplementary Table 10). Among these 30 mutations, 21 were homozygous and occurred only in *T. poliocephalus*, while the other nine mutations were either heterozygous in some other limestone langurs, homozygously present in the distantly related members of the *T. obscurus* group or, as in one case, heterozygous in one of the four *T. poliocephalus* individuals (Supplementary Data 8). To further investigate the potential adaptation of *T. poliocephalus* to saltwater consumption, we examined genes known to be involved in adaptation to different salinities in other vertebrates[41–51] and found in one gene (*CDH26*) a premature stop codon and in ten genes (*ASH1L, DPP10, CFAP65, COL14A1, COL17A1, EPPK1, PKP1, SLC4A9, SLC22A18, STAC*) at least one non-synonymous variant that occurred only in *T. poliocephalus* and not in any of the other *Trachypithecus* species (Supplementary Data 9).

## Discussion

### Phylogenetic relationships among northern limestone langurs

*Trachypithecus poliocephalus* is a member of the northern limestone langur clade, but the phylogenetic relationships among the three species in this clade remained disputed. While similarities in fur coloration suggest a sister group relationship between *T. poliocephalus* and *T. leucocephalus*[35], mitochondrial genome data supported a basal position of *T. poliocephalus* among northern limestone langurs[34]. However, mitochondrial DNA represents just a single locus and can result in branching patterns different from the species tree and nuclear phylogenies[52]. In our nuclear-based phylogenies (Fig. 1c; Supplementary Figs. 1 and 2), underpinned by the results of Admixture plots and PCA (Supplementary Figs. 4 and 5, Supplementary Table 5), *T. poliocephalus* is revealed as sister species to *T. francoisi* and *T. leucocephalus*. Overall, this supports the classification of the three northern limestone langur taxa as distinct species[31–34] and disagrees with the classification of these three taxa into only two species, *T. francoisi*, and *T. poliocephalus*, with the latter containing *T. leucocephalus* as a subspecies[35]. Further support for the species-level classification of the three northern limestone langur taxa is provided by the fact that they diverged approximately 0.9–1.0 Mya (Fig. 1c; Supplementary Fig. 3, Supplementary Table 4) and that no notable secondary gene flow among them was detected (Supplementary Tables 5 and 6). At least for *T. poliocephalus*, gene flow with its conspecifics would have been surprising as this species is endemic to Cat Ba Island and geographically clearly isolated from the other two species. Other islands in Ha Long Bay are either too small or do not contain suitable habitat, and on the mainland close to Cat Ba Island, limestone formations as potential habitat for limestone langurs are absent, suggesting that *T. poliocephalus* has been restricted to Cat Ba Island for an extended period of time[30].

## Genetic diversity and inbreeding

With π = 0.033%, *T. poliocephalus* showed one of the lowest nucleotide diversities among a set of 55 mammal species (Fig. 2a). Similarly, genome-wide autosomal heterozygosity was with $He = 0.36$ per 1000 bp (Fig. 2b) amongst the lowest documented for primates so far ($He = 0.34-7.14$)[53,54]. Although snub-nosed monkeys (*Rhinopithecus* spp.) have similarly low heterozygosity values as *T. poliocephalus* ($He = 0.34-0.42$; exception *R. brelichi*: $He = 0.69$)[53], other primates, such as the other limestone langur species ($He = 0.62-1.33$; Fig. 2b) or great apes ($He = \sim 0.6-2.4$)[28,55] showed a ~2–6-fold higher heterozygosity. Likewise, when compared to other mammals, for instance, some *Capra* species[18], Sumatran rhinoceros[20], mainland gray fox[13], European gray wolf[56], giant panda[57] or European brown bear[58] ($He$ values of ~0.77 to ~1.90; for details see Supplementary Data 10), *T. poliocephalus* exhibits a comparatively low heterozygosity. However, values were similar or higher than in Alpine ibex[18], moose (Minnesota population)[59], beluga[60], narwhal[60], vaquita[61], polar bear[62], brown hyena[63], snow leopard[64], cheetah[65], Iberian lynx[66], Eurasian lynx[66], and the San Nicolas population of the Channel Island fox[13] ($He$ values of 0.01 to 0.34; Supplementary Data 10). Many of these species have larger populations and/or distributions. Thus, considering that *T. poliocpehalus* is endemic to the relatively small island of Cat Ba (~140 km²) and that the current population traces back to only 40 individuals in 2002–2004, the observed heterozygosity of $He = 0.36$ is comparatively high. This suggests that the long-term population size was likely larger and only recently dropped to the low number of around 100 individuals estimated in the first survey, conducted in 1999[30].

In all six investigated limestone langur species, we observed a lower heterozygosity in protein-coding compared to non-protein-coding regions, which is in agreement with the general expectation that heterozygosity is lower in functionally important regions (Supplementary Fig. 6; Supplementary Data 1). However, in *T. poliocephalus* heterozygosity in protein-coding regions and the ratio of non-synonymous to synonymous variants were both higher (1.1–1.3-fold and 1.3–1.8-fold, respectively) than in its conspecifics (Supplementary Figs. 6 and 7; Supplementary Data 1). This is in line with predictions that in small populations, the proportion of non-synonymous variants can increase due to weakened selection which was previously shown for non-African humans[67,68] and the San Nicolas population of the Channel Island fox[13]. A similar result was found for the narwhal and beluga with overall lower heterozygosity in coding versus non-coding regions, a pattern which was also less prominent in the narwhal, i.e., again in the species with overall lower genetic diversity. This finding of overall low levels of heterozygosity combined with little difference in diversity levels between coding and non-coding regions across the narwhal genome was interpreted as evidence that heterozygosity levels have reached a diversity stasis across the genome and that any decreases in genetic diversity might be problematic for the longer-term survival of the species[60]. The same may apply to the Cat Ba langur, suggesting that efforts should be undertaken to ensure no further erosion of genetic diversity in this species.

To estimate the degree of inbreeding, we analyzed ROHs. In *T. poliocephalus*, the fraction of the genome in ROHs >100 kb was with 88.98% (Fig. 2c) the highest among limestone langurs, and the species exhibited also the longest ROHs (25.48–40.16 Mb) and the largest number of long ROHs (>1 Mb; Supplementary Figs. 8–10). Such comparatively long ROHs are indicative for recent mating between closely related individuals, most likely during the past few generations[69]. Similarly, the genomic inbreeding coefficient $F$ based on ROHs was with $F_{ROH} = 0.85$ (Fig. 2d) the highest among limestone langurs and higher than in other vertebrates known to be also affected by high levels of inbreeding ($F_{ROH} = 0.05-0.79$)[15,21,28,53,61]. Population structure as a potential explanation for the high $F$ is unlikely as males are known to regularly move between sub-populations.

Interestingly, *T. poliocephalus* showed a significantly higher nucleotide diversity while the fraction of ROHs was significantly smaller on the human chromosome 19 ortholog compared to other autosomes (Fig. 2e; Supplementary Fig. 11). The same pattern was observed for *T. leucocephalus*, a species with also a relatively small population size. Some of the other *Trachypithecus* species with generally larger population sizes showed the same trend, especially *T. francoisi*, but for none of them, the differences between the human chromosome 19 ortholog and the remaining autosomes were significant. The human chromosome 19 is well known for its unusually high gene density with more than double the number of genes compared to the genome-wide average and 20 tandemly clustered gene families[70]. This pattern seems to be conserved among primates[71], indicating the biological and evolutionary significance of chromosome 19 and its orthologs in other primates. Overall, our findings suggest that *T. poliocephalus*, despite low genome-wide diversity and high inbreeding level, maintained relatively high genetic diversity in functionally important regions such as protein-coding genes and the generally gene-rich human chromosome 19 ortholog.

## Deleterious mutations and genetic load

Genetic load refers to the reduction in individual and mean population fitness due to the accumulation of deleterious mutations[72,73]. Genetic load can be divided into realized load (all sites where a deleterious allele is expressed, mainly sites that are homozygous for recessive deleterious alleles) and masked load (sites that are heterozygous where a recessive deleterious allele does not contribute to loss of fitness)[72]. We observed in *T. poliocephalus* a significantly higher realized and a (significantly) lower masked load compared to the other two northern limestone langur species (Supplementary Fig. 12), implying an accumulation of deleterious alleles in homozygous state in *T. poliocephalus*. Similar patterns were also found in the pink pigeon[74], the Scandinavian wolf population[75,76], the Florida panther[77], and the Scandinavian Arctic fox population[78], all of which represent populations that have suffered from severe bottlenecks followed by inbreeding in the relatively recent past. Likewise, when investigating the impact of deleterious mutations, we observed among northern limestone langur species no significant differences in the proportion of all (homozygous and heterozygous) high-impact deleterious mutations (Fig. 3a). However, when considering only homozygous deleterious mutations, the proportion in categories high, moderate and low was significantly higher in *T. poliocephalus* compared to the other two species (Fig. 3b). Our findings are in agreement with predictions that high levels of inbreeding can lead to increased homozygosity of recessive deleterious mutations, an effect observed in a number of small and isolated populations[19,72,79,80].

## Adaptation to high calcium intake

Limestone langurs, including *T. poliocephalus*, live in karst habitats, land formations with steep and tall cliffs formed by highly soluble and porous bedrock such as limestone and characterized by alkaline soil with poor nutrient content except minerals[81,82]. Because food plants and drinking water in karst habitats contain high concentrations of calcium and other minerals, limestone langurs have a naturally high calcium intake[37,83–87]. It was previously shown that limestone langurs (five species have been genomically investigated so far: *T. francoisi*, *T. leucocephalus*, *T. hatinhensis*, *T. laotum,* and *T. ebenus*) are adapted to high calcium intake in that they downregulate the calcium entry into the cell[37]. Most likely responsible for this adaptation are amino acid changes in seven positively selected genes (*CACNA1B*, *CACNA1C*, *CD38*, *EGFR*, *HTR2B*, *ITPKB*, *MYLK*) of the KEGG "calcium signaling pathway" and "oxytocin signaling pathway"[37]. We found the same amino acid changes in *T. poliocephalus*, but also additional variants that may have further increased the species' tolerance to high calcium intake. Using whole-genome scans, we identified various genes related

to calcium metabolism that contain non-synonymous variants largely private to *T. poliocephalus* (Supplementary Fig. 14, Supplementary Table 9, Supplementary Data 7). Most of these genes encode proteins located on the cell membrane and are involved in ion binding and transfer. For instance, *MCTP2* (multiple C2 and transmembrane domain-containing protein 2), an integral component of the membrane, enables calcium ion and lipid binding[88,89]. Some of the other genes such as *ADCY1* (adenylate cyclase 1), *ASPH* (aspartate ß-hydroxylase), *P2RX6* (purinergic receptor P2X 6), *PLCD3* (phospholipase C delta 3), *RYR1* (ryanodine receptor 1), and *MYLK4* (myosin light chain kinase family member 4) are also known to play important roles in calcium homeostasis[37,90–96], and amino acid changes in these and other genes may have improved the adaptability of *T. poliocephalus* to high calcium intake even further.

## Potential adaptation to saltwater consumption

For most vertebrates, high external salt concentrations can lead to toxic levels of ion accumulation and water loss within cells[97]. Vertebrates living in or close to the sea are exposed to high salt concentrations and hence, salinity is an important extrinsic factor affecting their ecology, evolution, and distribution[98–100]. For some species, it is known that they have evolved genetic mechanisms to persist under salinized or brackish conditions[101]. *Trachypithecus poliocephalus* as a species living on a maritime island is exposed to saltwater in the form of moisture on food plants and animals are known to lick and drink brackish water[38], a behavior unique among primates. Thus, we hypothesized that *T. poliocephalus* may has also evolved mechanisms to cope with high salt concentrations.

We identified in *T. poliocephalus* 30 largely species-specific non-synonymous variants in 22 genes related to sodium transport and homeostasis (Supplementary Data 8, Supplementary Table 10). Previous studies showed that transmembrane transporters encoded by genes of the solute carrier family (SLC) have been linked to osmoregulation and salinity adaptation[45,46,50,51]. For instance, *SLC4A11* (solute carrier family 4 member 11) encodes a voltage-regulated, electrogenic sodium-coupled borate cotransporter, which mediates transcellular chloride ion reabsorption via SLC4A11 anion exchangers[102,103]. Zhu et al.[104] reported that salt-sensitivity of mice is associated with increased renal protein expressions of *SLC4A4* and common variants in *SLC4A4* contribute to variation in blood pressure responses to dietary sodium intake in Han Chinese[105]. In both genes, *T. poliocephalus* exhibits unique amino acid changes, and other genes with unique or largely unique amino acid changes in *T. poliocephalus*, such as *SLC5A1*, *SLC5A2*, *SCNN1D*, and *SCN5A*, are also known to play important roles in sodium ion transportation[106–108]. Using a list of genes potentially related to adaptation to different salinities in vertebrates[41–51], we identified another ten genes (*ASH1L*, *DPP10*, *CFAP65*, *COL14A1*, *COL17A1*, *EPPK1*, *PKP1*, *SLC4A9*, *SLC22A18*, *STAC*) which contained species-specific non-synonymous variants in *T. poliocephalus* (Supplementary Data 9). All these genes are involved in ion transportation, osmoregulation, homeostasis, and cell-cell adhesion[41–51].

Another gene, potentially related to adaptation to different salinities, is *CDH26* (cadherin 26)[42,43,48]. All four *T. poliocephalus* individuals exhibited in *CDH26* a premature stop codon resulting in the loss of the complete cytoplasmatic component (Supplementary Figs. 15–17). CDH26 is a member of the cadherin protein family, which are calcium-dependent adhesion molecules that mediate cell-cell adhesion in all solid tissues and modulate a wide variety of processes including cell polarization, migration, and differentiation[109–112]. In humans, *CDH26* is known to exhibit high expression levels in prostate and urinary bladder[113]. Moreover, over-expression of *CDH26* might be related to myocardial infarction and progression of atherosclerosis[114,115], and high salt diet is known as high-risk factor for cardiovascular disease which creates a substrate for arrhythmias, myocardial infarction, and atherosclerosis[116–119]. In the American green treefrog, lower expression

of *CDH26* in coastal versus inland populations was suggested as adaptation to higher salinity[42,43] and a genomic study of vendace showed that outlier single nucleotide polymorphisms (SNPs) in *CDH26* may be associated with divergent selection related to environments exhibiting different salinities[48]. Thus, down-regulated expression of *CDH26* might be an important factor for high salt adaptation in vertebrates and we speculate that the disrupted protein function (loss of the intracellular catenin-interacting domain) may have contributed, in combination with amino acid changes in genes related to sodium metabolism, to genomic adaptation to increased saltwater tolerance in *T. poliocephalus*.

Our whole-genome analysis of the Cat Ba langur illustrates the competing effects, small population size has on genetic diversity. While our data reveal low genetic diversity across the genome as well as long runs of homozygosity and an accumulation of deleterious mutations, genetic diversity has been partially preserved in functionally important regions. Our study also revealed the potential genetic basis of adaptations of this species to its unusual insular habitat, particularly to high calcium intake and saltwater consumption. However, these results need to be treated as preliminary because only four Cat Ba langur individuals have been investigated in this study, albeit this refers to approximately 5% of the species' global population. Undoubtedly, the Cat Ba langur is unique among primates and even among limestone langurs, further emphasizing the importance to protect this critically endangered species.

## Methods
### Sample collection
We obtained blood samples from two translocated females and tissue samples from two deceased infants from the Cat Ba Langur Conservation Project. The blood samples were taken by experienced veterinarians in 2012 during a wild-to-wild translocation aimed at reintroducing two isolated females to the larger of the two breeding sub-populations. The translocation was initially proposed prior to 2008, but the master plan was submitted to relevant Vietnamese authorities and the international community in 2010. Approval to carry out the translocation was granted by HPPC (No: 4398/UBND-NN) and MARD (No: 245/TCLN-BTTN) in early 2010. The two females were immobilized by the delivery of chemical agents (ketamine and medetomidine) via a blow dart after being trapped in a sleeping cave. After initial immobilization, they were lowered one by one to the ground by a basket and there intubated and maintained on gaseous anesthesia. Immobilized animals were continuously observed for vital parameters such as respiration, pulse frequency, and internal body temperature. Whole blood samples (5 ml) were collected from the femoral vein, placed in EDTA tubes, and kept frozen at −80 °C until DNA extraction. Tissue samples from the two deceased infants found in the wild were collected in 2015 and 2018 and stored in 80% ethanol until further processing. All research complied with protocols approved by the Animal Welfare Body of the German Primate Center and adhered to the legal requirements of Vietnam. We conducted the study in compliance with the Convention on International Trade in Endangered Species of Wild Fauna and Flora (CITES; export nr. 18VN0331N/CT-KL, import nr. DE-E-02092/18) and the principles of the American Society of Primatologists for the ethical treatment of non-human primates.

### DNA extraction and sequencing
DNA was extracted with the Gentra Puregene Blood & Tissue Kit (Qiagen) following the manufacturer's instructions. DNA quality was checked with pulsed-field gel electrophoresis and concentration was measured with a NanoDrop Microvolume Spectrophotometer (ThermoFisher). 200 ng of DNA was subjected to whole-genome sequencing following the Illumina DNA prep workflow. Sequencing was done on Illumina's HiSeq 4000 (151 bp paired-end) to a mean coverage of 32.7× (Supplementary Table 1) at Novogene China. Short-read sequencing

data of the four *T. poliocephalus* individuals are available on NCBI under BioProject PRJNA949813.

## Additional sequence data

For comparative reasons, we added sequence data of another 23 individuals representing nine *Trachypithecus* species (each eight individuals of *T. francoisi* and *T. leucocephalus*, and each one individual of *T. laotum*, *T. hatinhensis*, *T. ebenus*, *T. germaini*, *T. auratus*, *T. obscurus* and *T. phayrei*) which were obtained from a recently published study (BioProject PRJNA488530)[37]. Additional outgroup taxa were download from NCBI's Sequence Read Archive (SRA): *Rhinopithecus roxellana* (SRR8718596, SRR8718597), *Colobus angolensis* (SRR1687497), *Papio anubis* (SRR8762000) *Macaca mulatta* (SRR16119994), *Chlorocebus sabaeus* (SRR6196475), *Pongo abelii* (SRR11032814), *Gorilla gorilla* (SRR9703449), *Pan troglodytes* (SRR11892898) and *Homo sapiens* (SRR11075380). Published paired-end SRA data were split by sratoolkit v2.9.6 (https://trace.ncbi.nlm.nih.gov/Traces/sra/sra.cgi?view=software) using "fastq-dump --split-3" parameters following the NCBI protocol and then compressed with pigz v2.7 (http://zlib.net/pigz/) using multiple default threads.

## Mapping and genotype calling

Raw sequence reads were adapter-trimmed and quality-filtered with fastp v0.23.1[120] with 1. reads with unidentified nucleotides ($N$) > 10% discarded, and 2. reads with the proportion of low-quality base (phred quality < = 5) > 50% discarded. We then mapped high-quality reads to the reference genome of either *M. mulatta* (Mmul_10) or *T. francoisi* (Tfra_2.0, https://www.ncbi.nlm.nih.gov/genome/31690?genome_assembly_id=749809, GCF_009764315.1) using the Burrows-Wheeler Aligner v0.7.12[121] with MEM algorithms. SAM format files were imported to samtools v1.9[122] for sorting with recommend parameters and then imported to Picard v2.20.2 (http://broadinstitute.github.io/picard/) to remove duplicates and build indexed BAM files. Mapping results for the four *T. poliocephalus* samples to Tfra_2.0 are shown in Supplementary Tables 1 and 11. We genotyped reads using a pipeline implemented in GATK v4.2.2[123]. SNP calling was performed following GATK's best practice. This included realignment of insertion/deletion (indel) polymorphisms with the "RealignerTargetCreator" and "IndelRealigner" functions, which were used to re-calibrate quality scores, excluding from the error model variant positions that were pre-called using HaplotypeCaller. For the genotype calling, we obtained the GVCF file for each individual using the "HaplotypeCaller" method in GATK and then, using the GenotypeGVCFs based method with the "includeNonVariantSites" flag, to get the population VCF file, including all confident sites. We then applied the "SelectVariants" to exclude indels and split the data into variant and nonvariant sites. The hard filter command "VariantFiltration" was applied to exclude potential false positive variant calls with the following criteria: "filterExpression QD < 2.0 || FS > 60.0 || MQ < 40.0 || ReadPosRankSum < --8.0 || MQRankSum <12.5" and "genotypeFilterExpression DP < 4.0". Additionally, sites were removed if there was an "N" in the reference sequence or the site spanned an indel plus a buffer of 3 bp in both directions and the site included >10% missing genotypes. To obtain the genotype file for subsequent analyses, a PERL script was used to transfer the VCF format to genotype format (e.g., AA, AT) and degenerate bases format (e.g., 'M' = 'AC') for all langurs and then again to generate final genotypes in VCF format. Our final VCF files contained 89,241,059 and 53,321,193 variants derived from the mapping to Mmul_10 and Tfra_2.0, respectively. Following the snpEFF v4.3[124] best-practice pipeline for annotation, all individual genotype files were annotated using an own-build Tfra_2.0 dataset (both gtf and gff3 files were download from NCBI; Supplementary Table 1).

## Relatedness among individuals

To investigate the kinship coefficient among the four *T. poliocephalus* individuals, we used King v2.3.2[125] to check family relationship and flag pedigree errors (Supplementary Table 2). An estimated kinship coefficient range >0.354, [0.177, 0.354], [0.0884, 0.177] and [0.0442, 0.0884] corresponds to duplicate sample/monozygotic twin, 1st-degree, 2nd-degree, and 3rd-degree relationship, respectively[125].

## Phylogeny, population structure, and gene flow

Phylogenetic trees based on autosomal SNPs were calculated with NJ and ML algorithms using *R. roxellana* as outgroup. The NJ tree was generated with TreeBest software (http://treesoft.sourceforge.net/treebest.shtml) with 1000 bootstrap replicates and the HKY model as it was determined as the most appropriate model by MrModeltest[126]. The ML tree was reconstructed in IQ-tree v2.1.3[127] with 1000 ultrafast bootstrap replicates[128,129] and the 'TVM + F + R2' model as automatically determined by IQ-tree. FigTree v1.4.0 (http://tree.bio.ed.ac.uk/software/figtree/) was used to visualize phylogenetic trees. Divergence times were calculated in IQ-tree based on coding regions on autosomes (CDS regions were selected using Mmul_10 annotation gtf files) using the phylogenetic dating option[130] in IQ-tree and applying a relaxed clock model. For this analysis, the 'TVM + F + I + G4' model was selected as best-fit model and again 1000 ultrafast bootstrap replicates were performed. To obtain 95% confidence intervals we resampled branch lengths 100 times and used the default setting of 0.2 for the standard deviation of the lognormal distribution[130]. The final alignment with a length of 36,811,173 bp contained 36 sequences (27 *Trachypithecus* individuals, *R. roxellana*, *C. angolensis*, *M. mulatta*, *P. anubis*, *C. sabaeus*, *P. abelii*, *G. gorilla*, *P. troglodytes*, *H. sapiens*). To calibrate the molecular clock, we constrained eight nodes based on fossil evidence[131]. As IQ-tree requires at least one fixed node age and allows only to set hard minimum, hard maximum, or both, we constrained the eight nodes as follows: 1. most recent common ancestor (MRCA) *Homo* and *Pan* 4.631–15.0 Mya; 2. MRCA Hominidae 12.3–25.235 Mya; 3. MRCA *Papio* and *Macaca* 5.33–12.51 Mya; 4. MRCA Cercopithecinae 6.5–15.0 Mya; 5. MRCA Colobinae 8.1.25–15.0 Mya; 6. MRCA Cercopithecidae 12.47–25.235 Mya, and 7. MRCA Catarrhini 35.102 Mya[131].

The genetic population structure of *T. poliocephalus* and its closest relatives, *T. francoisi* and *T. leucocephalus*, was first inferred with *frappe* v1.1[132]. Admixture v1.3.0[133] was used to estimate individual ancestry proportions. We predefined the number of genetic clusters *K* from 2 to 5 and used cross-validation error methods to choose the best *K* value. The maximum iteration of the expectation-maximization algorithm was set to 10,000. The PCA was conducted with EIGENSOFT v7.2.1[134] and the significance of eigenvectors was determined with the Tracy-Widom test in EIGENSOFT.

We analyzed allele sharing using *D*-statistics, with qpDstat of the Admixtools package[135,136] and Dtrios & Dinvestigate programs in Dsuite[137] to explore potential gene flow events between *T. poliocephalus*, *T. francoisi* and *T. leucocephalus*. Using quartets of populations with the topology ((($P_1$, $P_2$), $P_3$), O), where O represents the outgroup (*R. roxellana* in our study), we computed the normalized product of the allele differences for population 1 ($P_1$) and 2 ($P_2$), and population 3 ($P_3$) and outgroup, averaged over all SNPs. By a whole-genome scan using qpDstat and Dtrios, we obtained the best model among these three species.

## Genetic diversity, runs of homozygosity and inbreeding

To infer the nucleotide diversity π of *T. poliocephalus* and compare it with that of other mammal species, we measured differences between chromosome pairs in the wild-born limestone langurs (rainforest langurs were excluded as three of them, *T. obscurus*, *T. germaini* and *T. auratus*, were captive born and thus may not reflect the nucleotide diversity characteristic for wild-born animals) with vcftools v0.1.14[138]

(genome-wide-site methods, --site-pi and non-overlapping-50kb-window methods, --window-pi 50000). Data from other mammal species were obtained from previous publications[18,139,140]. We counted the genome-wide autosomal heterozygosity and heterozygosity in protein-coding (including non-synonymous and synonymous variants) versus non-protein-coding regions for all individuals using genotype call files from merged VCF files and the ".gtf" annotation files from NCBI.

We inferred ROHs across the limestone langur genomes using PLINK v1.9[141] and bcftools v1.7[121]. To this end, we ran sliding windows of 20 SNPs on the VCF files of included genomes, requiring at least one SNP per 50 kb (parameters of PLINK: --homozyg --homozyg-density 50 --homozyg-gap 100 --homozyg-kb 100 --homozyg-snp 50 --homozyg-window-het 1 --homozyg-window-snp 20 --homozyg-window-threshold 0.05). In each individual genome, we allowed for a maximum of one heterozygous and 50 missing calls per window before we considered the ROH to be broken. As only for *T. poliocephalus*, *T. francoisi* and *T. leucocephalus* more than three individuals were available, we performed comparative ROH analyses on population level only for these three species, using One-way ANOVA test to check for significant differences. The inbreeding coefficient *F* for each species was calculated with PLINK v1.9 as (observed homozygous SNPs-expected homozygous SNPs)/(total called SNPs-expected homozygous SNPs) for each sample and then plotted separately for each species.

### Deleterious mutations and genetic load

We downloaded a multiple alignment with 100 vertebrate species ("100way alignment") including Mmul_10 from the UCSC genome browser (http://hgdownload.soe.ucsc.edu/goldenPath/hg38/multiz100 way/maf/). The GERP scores were subsequently transferred from hg38 to Mmul_10 using LiftOver[142]. The vcfR package was used to read the vcf files into r v4.1.1[143] and all subsequent calculations were performed in R using the package tidyverse[144]. After removing identical positions shared by all three limestone langur species, we used autosomal polymorphism data to estimate the individual masked load as the sum of the GERP scores of all deleterious derived alleles in heterozygous genotypes, divided by the number of called genotypes per individual to account for differences in callability between individuals. The realized load was calculated as the sum of GERP scores of deleterious derived alleles in homozygous genotypes divided by all called sites in the genome. Based on the distribution of synonymous and deleterious non-synonymous mutations, we set a GERP score threshold of 4 to define a mutation as potentially deleterious[18,76,140,145]. To calculate the proportion of heterozygous versus homozygous derived genotypes in an individual, we divided the number of sites of each genotype by the total number of called genotypes for that individual (including sites homozygous for the ancestral allele, genotyped because they were polymorphic in other individuals in our dataset)[76]. Obtained SNPs were then grouped into four effect categories (modifier, low, moderate, high) based on snpEFF annotation (a simple estimation of putative impact/deleteriousness) and compared among the three northern limestone langur species. For genes containing homozygous deleterious mutations of high impact we performed functional enrichment analysis with KOBAS 3.0[146,147]; corrected *p* values were calculated using Benjamini and Hochberg corrected test.

### Positive selection and non-synonymous variants

To identify genomic regions that may have been subject to selection in *T. poliocephalus*, we investigated pair-wise comparisons between *T. poliocephalus* and *T. francoisi*. We calculated average nucleotide diversity ($\theta\pi$), using the genome-wide sliding windows. As the variance of $\theta\pi$ depends on the number of SNPs used for each calculation, spurious selection signals will be more likely in windows with few variable sites. To reduce the number of false positives we used only windows with a minimum of 20 variable sites. We tested a range

of different window sizes (20, 40, and 100 kb) and note that a window size of 40 kb (20 kb steps) resulted in a low number of windows with few SNPs. Consequently, and also according to linkage distance analysis, a window size of 40 kb allowed us to screen a large fraction of the genome at a false positive rate that likely is lower than if a smaller window size would have been used. Then, we estimated the cross-population extended haplotype homozygosity (XP-EHH)[148] statistic with R package rehh[149]. The XP-EHH scores for all variants were also Z-transformed using the math module in python v2.7. The threshold for identifying candidate selective regions in the XP-EHH was set to the top 5% outliers. Furthermore, we used KaKs_calculator v2.0[150] to identify positively selected sites based on sliding windows across the full CDS sequence with "-YN" parameters. As our selection methods are pruned to small sample size and to avoid any bias, we recognized only genes under strong selective sweep when they were supported by at least two selection methods. The functional classification and enrichment analyses of candidate genes under selective sweep were performed with KOBAS v3.0[146,147]; corrected *p* values were calculated using the Benjamini and Hochberg corrected test.

We used snpEFF annotated VCF files and identified non-synonymous variants in *T. poliocephalus* following two criteria: 1. mutations must be classified as non-synonymous variants, and 2. these variants should be homozygous in all four *T. poliocephalus* individuals. Next, we investigated whether genes with these non-synonymous variants were supported by any of the three selection methods based on the top 5% outlier results (see above). We further screened genes known to be responsible for adaptation to high calcium intake in other limestone langur species[37] and performed literature searches to identify candidate genes potentially related to differential salinity adaptation in vertebrates[41–51]. Genes reported in these studies were then checked for the presence of non-synonymous mutations unique to *T. poliocephalus* using snpEFF.

### Reporting summary

Further information on research design is available in the Nature Portfolio Reporting Summary linked to this article.

## Data availability

Newly generated short-read sequencing data are available on NCBI under BioProject PRJNA949813 (https://www.ncbi.nlm.nih.gov/bioproject/949813). Publicly available genome data used in this study can be found under BioProject PRJNA488530 (https://www.ncbi.nlm.nih.gov/bioproject/488530) and in Sequence Read Archive SRR8718596, SRR8718597, SRR1687497, SRR8762000, SRR6196475, SRR16119994, SRR11032814, SRR9703449, SRR11892898, and SRR11075380. Samples of the two deceased Cat Ba langur individuals and the two translocated females are stored at the Cat Ba Langur Conservation Project, Cat Ba Island, Vietnam, and the Wildlife Conservation Society, Hanoi, Vietnam, respectively. Source data are provided with this paper.

## Code availability

We have described all the tools and methods used for the analyses in the "Methods" section.

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

## Acknowledgements

We thank Tran Van Lan, Nguyen Ba Tiep, Hoang Van Thap, Nguyen Van Phien, Vu Hong Van, Alexandra Kolodyazhnaya, and Michael Meyerhoff for logistic support and help with data analysis, and Stephan D. Nash for permission to use his langur illustrations. We are grateful to the Cat Ba langur Conservation Project, Zoo Leipzig, Allwetterzoo Münster, the Zoologische Gesellschaft für Arten- und Populationsschutz, the Endangered Primate Rescue Center, Cat Ba National Park, Cuc Phuong National Park and the Ministry for Agricultural and Rural Development for continuous support. Financial support for this research was provided by grants from the German Research Foundation (HO 3492/9-1 to M.H. and RO 3055/7-1 to C.R.), Sino-German Mobility Programme (M-0084 to M.Li and C.R.), Chinese Academy of Sciences (CAS XDB31000000 to M.Li) and National Natural Science Foundation of China (NSFC 31821001 to M.Li) and Vietnamese Ministry of Science and Technology's Program 562 (ĐTĐL.CN-64/19 to M.D.L.). T.M.-B. was supported by funding from the European Research Council (ERC) under the European Union's Horizon 2020 research and innovation program (grant agreement No. 864203), PID2021-126004NB-100 (MICIIN/FEDER, UE), "Unidad de Excelencia María de Maeztu", funded by the AEI (CEX2018-000792-M), NIH 1R01HG010898-01A1 and Secretaria d'Universitats i Recerca and CERCA Programme del Departament d'Economia i Coneixement de la Generalitat de Catalunya (GRC 2021 SGR 00177).

## Author contributions

L.Z., T.M.-B., M.H., M.Li, Z.L., and C.R. designed the research; L.Z., L.W., M.H., and C.R. analyzed the data; N.L., R.P., M.S.L., P.V.T., L.T.N.H., N.H.C., L.V., M.Ly., A.E.F., N.T.T.N., N.V.L., B.M.R., A.B., N.V.T., M.D.L., and T.N. collected samples; L.Z., N.L., M.H., and C.R. wrote and revised the paper with contributions from all authors.

## Competing interests

The authors declare no competing interests.

## Additional information

[1]Primate Genetics Laboratory, German Primate Center, Leibniz Institute for Primate Research, Göttingen, Germany. [2]International Max Planck Research School for Genome Science (IMPRS-GS), University of Göttingen, Göttingen, Germany. [3]CAS Key Laboratory of Animal Ecology and Conservation Biology, Institute of Zoology, Chinese Academy of Sciences, Beijing, China. [4]Cat Ba Langur Conservation Project (CBLCP), Cat Ba National Park, Cat Ba Island, Cat Hai District, Hai Phong Province, Vietnam. [5]Taronga Conservation Society Australia, Mosman, NSW, Australia. [6]Melbourne Zoo, Zoos Victoria, Parkville, VIC, Australia. [7]Wildlife Conservation Society (WCS), Health Program, New York, NY, USA. [8]Wildlife Conservation Society (WCS), Vietnam Country Program, Hanoi, Vietnam. [9]World Wildlife Fund for Nature (WWF) International, Gland, Switzerland. [10]School of Archaeology and Anthropology, The Australian National University, Canberra, ACT, Australia. [11]Evolutionary Adaptive Genomics, Institute of Biochemistry and Biology, Department of Science, University of Potsdam, Potsdam, Germany.

[12]Central Institute for Natural Resources and Environmental Studies, Vietnam National University, Hanoi, Vietnam. [13]Faculty of Environmental Sciences, University of Science, Vietnam National University, Hanoi, Vietnam. [14]Three Monkeys Wildlife Conservancy, Nho Quan District, Ninh Binh Province, Ninh Binh, Vietnam. [15]Institute of Evolutionary Biology (UPF-CSIC), PRBB, Barcelona, Spain. [16]Catalan Institution of Research and Advanced Studies (ICREA), Barcelona, Spain. [17]CNAG-CRG, Centre for Genomic Regulation (CRG), Barcelona Institute of Science and Technology (BIST), Barcelona, Spain. [18]Institut Català de Paleontologia Miquel Crusafont, Universitat Autònoma de Barcelona, Edifici ICTA-ICP, Cerdanyola del Vallès, Spain. [19]College of Life Sciences, Capital Normal University, Beijing, China. [20]Gene Bank of Primates, German Primate Center, Leibniz Institute for Primate Research, Göttingen, Germany. ✉e-mail: lzhang@dpz.eu; michael.hofreiter@uni-potsdam.de; lim@ioz.ac.cn; liuzj6888@cnu.edu.cn; croos@dpz.eu

