## [Peer Review File · Nature Communications]

Genomic adaptation to small population size and saltwater consumption in the critically endangered Cat Ba langurReviewers' Comments:

Reviewer #1:

Remarks to the Author:

The focus of the manuscript was to explore genetic diversity and genomic adaptation of a critically endangered langur species, the Cat Ba langur (*Trachypithecus poliocephalus*). To do so the authors carried out whole genome resequencing of four individuals. The authors combined previous sequenced individuals of two closely related langur species (*T. francoisi* and *T. leucocephalus*), and employed a series of population and conservation genomic analyses. The authors were able to obtain divergence times, population structure, gene flow among the three species. Moreover, they were able to estimate inbreeding level through analyses of ROH and genetic load in order to explore the genetic legacy of an endangered species. The most interesting part of the study is that they employed several analyses to detect candidate genes involved in Cat Ba langur's adaptive adaptation, such as high calcium intake, high salt water consumption and climbing movement. Overall, the authors define a picture of the evolution of an endangered primate species.

General Comments:

Overall, the article includes a considerable amount of analyses that is brought to bear on how a critically endangered primate evolve in an unique landscape. The article is a valuable contribution to our understanding of conservation genomics and adaptive evolution of extremely small population. However, I feel the two main themes of the study, i.e. genomic legacy of a small population and local adaptation is not well connected. Some of analyses (i.e. Population structure, PCA and D-statistics) are not necessary to support these two main themes. Justifications of doing these are missing. For the conservation genomic session, I wonder whether the authors are able to separate the overall genetic load to realized and masked load, separately, whether they can annotated the function of deleterious mutations in order to determine potential risk of the population. The functional analysis of detecting candidate focused on very specific life styles and behaviour of this species, which were already targeted by the authors in their previous studies. I'm wondering whether there are other selective regions involved the divergence of the species. The vast majority of my comments are attempts to improve clarity in parts of the manuscript. I would encourage the authors to rewrite portions of the Results and Discussion, especially the Results part, which are mixed with methods, results and interpretation of results, making it is a bit difficult to follow. Many of the ideas in the Results are valuable, but these ideas are not always described well and in many cases the ideas are not connected to each other well. A focused revision of the discussion would considerably improve the manuscript.

Specific Comments

Abstract:

Line 64: Is chromosome 19 unique in all primates, or just in langur species?

Introduction :

Line 89: Would be nice to show a photo of the study species in Figure 1 in order to make readers to see the species.

The introduction is very short. A nice summary of existing knowledge of the evolutionary history of the three langur species will be helpful. Whether their distributions are overlapped (at least historically), which may lead to the likelihood of historical introgression among them. I think these information is useful to formulate scientific questions.

L 106-114: Better to propose specific research questions here based on previous studies.

Results:

L 117-153: Because of lacking justifications and background information in introduction, I feel some analyses of these parts are not necessary. For instance, why PCA and admixture are useful?

L 154-173: To my opinion, the recent demographic history is more important for the study species. PSMC definitely has no power to illustrate this. I suggest the authors used more methods to approve it. For instance, SMC++ and GONE may be suitable methods. Further the divergence models among the three species can be inferred in more sophisticated approaches, such as fastsimcoal. However, the small sample size of the study might limit the use of this modelling approach.

L 167: Remove “individuals” as N_e has no unit.

L 175-188: The whole part is very descriptive. Authors may summarize these comparisons.

L 189-200: Would be nice to see the correlation between the length of ROH and number of ROH, because it can be also suggestive of historical bottleneck.

L 203: I do not find any discussion about the ‘magic chromosome 19’. Why it is special?

L 241: Would it be possible to know the function of genes containing deleterious mutations?

L 255: I feel the tests of exploring genes under selection are robust as authors used six different methods. As I said in the general comments, these results were targeted to calcium intake adaptation, saltwater tolerance, climbing adaptation. Whether other interesting genes were detected?

L 365: This part is simply the Conclusion of the study. Not Discussion per se.

Methods:

L 423-425: Both mapping protocols using different reference genomes were employed to admixture analyses? I’m a bit confused here.

L 547: Beside F_{st} and Tajima’s D , d_{xy} and local recombination rate might be other choice of measurement to detect signal of selection. Did authors consider to combine them into the analysis?

Reviewer #2:

Remarks to the Author:

This is a review of the manuscript: “Genomic adaptation to small population size and saltwater consumption in the critically 2 endangered Cat Ba langur” by Zhang et al. The study performs population genetic analyses of the endangered Cat Ba langur. It is highly relevant to perform a population genetic study on this species, given its precarious conservation status and unique biology. As such the study is timely, but I found several analytical concerns that need to be addressed for this manuscript to become ready for publication, in my opinion. Some of these lead to conclusions that are either unexpected or unreliable, given the limitations of the methods

applied.

ROH estimation, inbreeding and diversity:

I am skeptical that the authors have been able to accurately infer ROHs, particularly their length distributions. Given the authors finding of a continuously declining population size, I see no obvious explanation for why most ROHs should be short (100kb-1mb). I think the authors should reconsider whether they are using too strict criteria for breaking up ROHs, and therefore erroneously inferring many short rather than fewer long ROHs.

This observation is supported by the large inferred inbreeding coefficient – if F is really 0.85 as calculated in Fig. 2d using observed and expected het, then there should be LOTS of long ROHs. Another possible explanation for this excessively high F is, however, population structure, and I think the authors need to address whether this could be a problem for their analyses (see below). Perhaps this can also be affected by mapping to a distant reference, but I could not keep track of whether the ROH estimation was based on mapping to the close or to the distant reference genome. If the latter, it is almost certain that many spurious hets will be present to break up the ROHs into artificially short ROHs. One way to address the accuracy of inferred ROHs and their length is to simply plot them along the chromosomes and see if some putative longer ROHs are very visually being chopped up by the Plink ROH identification criteria.

I have a hard time understanding the authors' claim that heterozygosity is 30-80 times higher in coding than in non-coding regions for the langurs. This is totally unexpected, regardless of any selection scenario that I can imagine. I tried to follow the authors calculation provided in Table S10, but this did not make any sense to me in relation to what is described in the methods. For example, I do not understand what the fixed numerator in the final two columns of Table S10 are. Also, I don't understand why some individuals have very different numbers of heterozygous calls than their conspecific individuals. E.g. *T. poliocephalus* 1 has nearly 5 times as many heterozygous sites as *T. poliocephalus* 3. These results and calculations are not explained anywhere, so I found it hard to follow the authors' calculation of heterozygosity, both in general and inside/outside coding regions. In relation to the above, it was unclear to me why the authors used ANGSD and realSFS for this analysis, as they have high-depth data and genotype calls from a GATK pipeline. As I understand the authors, they obtained a vcf file including genotype calls in nonvariant sites, which could have been used for the above analysis.

Demographic analyses:

I don't think it is prudent to use e.g. stairway plots (or other, purely SFS based analyses) to draw conclusions about demographic history on such a short scale as the authors do. I do not believe that SFSes can truly inform us about events within the last 100 years or even decades, as in Fig. 1e. At least I have never seen this before, and many papers highlight the poor performance of Stairway plots (and similar methods) in the very recent and very remote past, especially when sample sizes are so tiny. In the same vein, I think the authors need to show the underlying SFSes, because there could be issues related to e.g. population structure. The authors highlight that these samples are taken from different localities ("subpopulations"), and these demographic analyses are all super sensitive to population structure.

Genetic load:

The authors claim that the lower ratio of homozygous LoF/synonymous in Cat Ba langurs can be explained by purging (L248), but I don't think this is what would happen by purging – purging would

perhaps (although this is not necessarily true) lead to a lower frequency of LoF variants relative to some other types of variants because recessive LoF are being removed by selection, but it should not lead to fewer HOMOZYGOUS LoF. Furthermore, the authors do not find a lower frequency of High impact deleterious alleles in Cat Ba langurs than other species (Fig. S15), which is not consistent with accelerated purging in this population.

In Table S12 I noticed that the depth of the homozygous LoF variants vary a lot even when looking across sites within the same individual. For example, DP for Tpol1 ranges between 5-128. This to me is a potential sign of problems, and I noticed that the authors only imposed a depth filter of minimum 4 when calling genotypes (L435). I would suggest choosing a narrower depth range for filtering SNPs, as very low or very high coverage regions potentially arise from mapping problems.

Positive selection and adaptation:

I would argue that the sample sizes are problematically low for some of the scan statistics that the authors apply. Furthermore, given the considerable evolutionary distance to the comparison population, some measures (certainly F_{st}) seem like an odd choice to infer positive selection. I am assuming that F_{st} genome-wide between the two langur species is close to 1. Outlier scans based on statistics with very narrow distributions will not work very well, and will be strongly affected by noise e.g. related to low sample sizes. If the inbreeding coefficient is really as high as estimated by the authors, this will lead to many spurious windows of low genetic diversity, making this a poor scan statistic as well (in addition to the effect of small sample size). I wonder why the authors did not use standard measures from comparative genomics, such as dN/dS ratios, to infer positive selection in the Cat Ba langur, given the low sample sizes and long evolutionary distances to the comparison populations? The process of taking the overlap of outliers from different methods is pretty ad hoc, and it should be acknowledged that the false-positive rate of such analyses is unknown (top 5% is quite inclusive).

Some of the analyses performed to identify unique adaptations are rather ad hoc or informal. For example, the finding of a single fixed LoF variant in gene CDH26 is taken as putative evidence of positive selection. While this is possible, there are other evolutionary processes that could lead to the same result, e.g. inbreeding (which the authors have reported to be very high) or lack of sufficiently efficient negative selection. In fact, in the genetic load section of the study, the authors implicitly assume that LoF variants are under negative selection. The authors should discuss these limitations, and/or supplement them with more formal statistical tests for positive selection. In addition, the functional role of this mutation as an adaptation to saltwater intake is speculative and should be presented as such.

In a similar vein, the results presented as indicative of “Enhanced climbing ability” are extremely speculative. Either the authors should remove this section, temper their interpretation substantially or substantiate their conclusion by additional experiments or data.

Reviewer #1 (Remarks to the Author):

The focus of the manuscript was to explore genetic diversity and genomic adaptation of a critically endangered langur species, the Cat Ba langur (*Trachypithecus poliocephalus*). To do so the authors carried out whole genome resequencing of four individuals. The authors combined previously sequenced individuals of two closely related langur species (*T. francoisi* and *T. leucocephalus*), and employed a series of population and conservation genomic analyses. The authors were able to obtain divergence times, population structure, gene flow among the three species. Moreover, they were able to estimate inbreeding level through analyses of ROH and genetic load in order to explore the genetic legacy of an endangered species. The most interesting part of the study is that they employed several analyses to detect candidate genes involved in Cat Ba langur's adaptive adaptation, such as high calcium intake, high salt water consumption and climbing movement. Overall, the authors define a picture of the evolution of an endangered primate species.

General Comments:

Overall, the article includes a considerable amount of analyses that is brought to bear on how a critically endangered primate evolves in a unique landscape. The article is a valuable contribution to our understanding of conservation genomics and adaptive evolution of extremely small populations.

However, I feel the two main themes of the study, i.e. genomic legacy of a small population and local adaptation, is not well connected.

Reply: Thanks for your overall positive evaluation of our manuscript and helpful comments. We rewrote large parts of the Introduction and now give more details about the conservation status (small population size), classification (disputed phylogenetic position) and biology (limestone habitat and saltwater drinking) of Cat Ba langurs, emphasizing the importance of understanding the genomic legacy of small populations and its relevance for environmental adaptation. In the Results and Discussion we further show how our genomic findings directly contribute to environmental adaptation. We hope that the three main topics of the study are better connected now – in the Introduction as well as in the Results and Discussion.

Some of the analyses (i.e. Population structure, PCA and D-statistics) are not necessary to support these two main themes. Justifications for doing these are missing. For the conservation genomic section, I wonder whether the authors are able to separate the overall genetic load into realized and masked load, separately, whether they can annotate the function of deleterious mutations in order to determine the potential risk of the population.

Reply: We expanded the Introduction, connecting the topics better and making clear why also the phylogeny/population structure part is needed (the phylogenetic relationships among northern limestone langurs are still disputed). However, we largely shortened the Results subsection "Phylogeny

and population structure”, changed Figure 1 by showing now a Cat Ba langur, sampling sites and only a dated phylogeny, and moved other figure panels from previous Figure 1 to the Supplementary Information.

We now calculated realized and masked loads (Supplementary Fig. 12) and discuss them in the Discussion subsection “Deleterious mutations and genetic load”. We further give a detailed list of all homozygous high-impact deleterious variants and performed enrichment analyses on those (Supplementary Tables 10 and 11).

The functional analysis of detecting candidate focused on very specific life styles and behaviour of this species, which were already targeted by the authors in their previous studies. I’m wondering whether there are other selective regions involved the divergence of the species. The vast majority of my comments are attempts to improve clarity in parts of the manuscript.

Reply: We highly appreciate your comment. We found in Cat Ba langurs a total of 232 genes under strong selection and functional enrichment analyses revealed that many of these genes are related to, among others, ATP and ribonucleotide binding, (metal) ion binding, signal transduction, calcium and oxytocin signaling pathways, diverse other regulatory cellular and metabolic processes, and diseases (e.g., blood pressure, obesity, nervous system, dental caries) (see Supplementary Table 13). We provide this information now in the first paragraph of the Results subsection “Positive selection and non-synonymous variants”. However, we make also clear that in our in-depth analysis, we focused on genes potentially associated with adaptation to the species’ unique environment characterized by high concentrations of minerals such as calcium and sodium.

I would encourage the authors to rewrite portions of the Results and Discussion, especially the Results part, which are mixed with methods, results and interpretation of results, making it is a bit difficult to follow. Many of the ideas in the Results are valuable, but these ideas are not always described well and in many cases the ideas are not connected to each other well. A focused revision of the discussion would considerably improve the manuscript.

Reply: Thanks for this helpful comment. We reorganized and re-wrote the Results and Discussion and present them now separately. The methods and interpretations are largely removed from the Results and our findings are then more deeply discussed in the Discussion. Moreover, we split the Discussion into different subsections with the aim to better synthesize and connect the ideas presented in the Results and to provide a more comprehensive and coherent interpretation of our results.

Specific Comments

Abstract:

Line 64: Is chromosome 19 unique in all primates, or just in langur species?

Reply: Human chromosome 19 is well known for its unusually high gene density with more than double the number of genes compared to the genome-wide average and 20 tandemly clustered gene families; this pattern seems to be conserved among primates. In the Discussion subsection “Genetic diversity and inbreeding” we provide more details about the uniqueness of human chromosome 19 and its orthologs in other primates.

Introduction:

Line 89: Would be nice to show a photo of the study species in Figure 1 in order to make readers to see the species.

Reply: We re-organized Figure 1 and present in Figure 1a now a Cat Ba langur licking saltwater from his tail.

The introduction is very short. A nice summary of existing knowledge of the evolutionary history of the three langur species will be helpful. Whether their distributions are overlapped (at least historically), which may lead to the likelihood of historical introgression among them. I think these information is useful to formulate scientific questions.

Reply: We expanded the Introduction and give more details about the three northern limestone langur species, but focused on Cat Ba langurs as this is our study species. In the last paragraph of the Introduction, we describe in more detail the aims of our study. We do not provide information about the distribution of the three northern limestone langurs in the Introduction, but discuss this topic at the end of the Discussion subsection “Phylogenetic relationships among northern limestone langurs”.

L 106-114: Better to propose specific research questions here based on previous studies.

Reply: We expanded the Introduction and present now specific research questions. In the last paragraph of the Introduction, we summarize the aims of the study and what was done.

Results:

L 117-153: Because of lacking justifications and background information in introduction, I feel some analyses of these parts are not necessary. For instance, why PCA and admixture are useful?

Reply: See reply above. We expanded the Introduction, connecting the topics better and making clear why also the phylogeny/population structure part is needed (phylogenetic relationships among northern

limestone langurs are still disputed). However, we largely shortened the Results subsection “Phylogeny and population structure”, changed Figure 1 by showing now a Cat Ba langur, sampling sites and only a dated phylogeny, and moved other figure panels from previous Figure 1 to Supplementary Information. The PCA and Admixture plots support tree-based findings that Cat Ba langurs are basal among northern limestone langurs (see Discussion subsection “Phylogenetic relationships among northern limestone langurs”).

L 154-173: To my opinion, the recent demographic history is more important for the study species. PSMC definitely has no power to illustrate this. I suggest the authors used more methods to approve it. For instance, SMC++ and GONE may be suitable methods. Further the divergence models among the three species can be inferred in more sophisticated approaches, such as fastsimcoal. However, the small sample size of the study might limit the use of this modelling approach.

Reply: We fully agree with the reviewer. Indeed, small sample numbers (and population structure) can lead to unreliable results, but other methods such as SMC++, GONE and fastsimcoal2 have similar problems. We applied these tools as well, but obtained unreliable results. As Reviewer 2 also doubted the reliability of the results, we decided to remove the demographic part from the manuscript completely.

L 167: Remove “individuals” as N_e has no unit.

Reply: See reply before. The whole demographic history part was removed.

L 175-188: The whole part is very descriptive. Authors may summarize these comparisons.

Reply: We agree and re-wrote/shortened this section.

L 189-200: Would be nice to see the correlation between the length of ROH and number of ROH, because it can be also suggestive of historical bottleneck.

Reply: Thanks for this advice. We calculated the length and number of ROHs. The results are presented in Results subsection “Genetic diversity and inbreeding”, shown in Supplementary Figs. 8–10, and discussed in Discussion subsection “Genetic diversity and inbreeding”.

L 203: I do not find any discussion about the ‘magic chromosome 19’. Why it is special?

Reply: See reply above. Human chromosome 19 is well known for its unusually high gene density with more than double the number of genes compared to the genome-wide average and 20 tandemly clustered gene families; this pattern seems to be conserved among primates. In the Discussion subsection “Genetic diversity and inbreeding” we provide more details about the uniqueness of human chromosome 19 and its orthologs in other primates.

L 241: Would it be possible to know the function of genes containing deleterious mutations?

Reply: See reply above. We now provide a list of all homozygous high-impact deleterious variants and performed enrichment analyses on those genes (Supplementary Tables 10 and 11).

L 255: I feel the tests of exploring genes under selection are robust as authors used six different methods. As I said in the general comments, these results were targeted to calcium intake adaptation, saltwater tolerance, climbing adaptation. Whether other interesting genes were detected?

Reply: See reply above. We found in Cat Ba langurs a total of 232 genes under strong selection and functional enrichment analyses revealed that many of these genes are related to, among others, ATP and ribonucleotide binding, (metal) ion binding, signal transduction, calcium and oxytocin signaling pathways, diverse other regulatory cellular and metabolic processes, and diseases (e.g., blood pressure, obesity, nervous system, dental caries) (see Supplementary Table 13). We provide this information now in the first paragraph of the Results subsection “Positive selection and non-synonymous variants”. However, we make also clear that in our in-depth analysis, we focused on genes potentially associated with adaptation to the species’ unique environment characterized by high concentrations of minerals such as calcium and sodium.

L 365: This part is simply the Conclusion of the study. Not Discussion per se.

Reply: We split now the previously combined Results and Discussion into separate sections. The previous Discussion is now the Conclusion.

Methods:

L 423-425: Both mapping protocols using different reference genomes were employed to admixture analyses? I’m a bit confused here.

Reply: We apologize for any confusion. In the Results, we added now a subsection “Sampling and datasets” in which we explain in detail which mapping data were used for which analysis. For simplification, we use for all phylogenetic and population genetic analyses now only the Mmul_10 data.

L 547: Beside F_{st} and Tajima's D , d_{xy} and local recombination rate might be other choice of measurement to detect signal of selection. Did authors consider to combine them into the analysis?

Reply: Thanks for your suggestion. We used already five methods, including F_{ST} based on divergence, Tajima D based on neutral theory, H_p based on heterozygosity, and XP-EHH and iHs based on haplotype, but not local recombination rate. We believe that the combination of five methods is already an exhausting way to detect strong selection.

Reviewer #2 (Remarks to the Author):

This is a review of the manuscript: "Genomic adaptation to small population size and saltwater consumption in the critically endangered Cat Ba langur" by Zhang et al. The study performs population genetic analyses of the endangered Cat Ba langur. It is highly relevant to perform a population genetic study on this species, given its precarious conservation status and unique biology. As such the study is timely, but I found several analytical concerns that need to be addressed for this manuscript to become ready for publication, in my opinion. Some of these lead to conclusions that are either unexpected or unreliable, given the limitations of the methods applied.

ROH estimation, inbreeding and diversity:

I am skeptical that the authors have been able to accurately infer ROHs, particularly their length distributions. Given the authors finding of a continuously declining population size, I see no obvious explanation for why most ROHs should be short (100kb-1mb). I think the authors should reconsider whether they are using too strict criteria for breaking up ROHs, and therefore erroneously inferring many short rather than fewer long ROHs.

Reply: Thank you for your insightful comments on our ROH analysis. Considering also the comments by Reviewer 1, we re-analyzed ROHs, using more relaxed criteria to identify these regions (see Methods). We now also provide length and number of ROHs as well as the genome fraction in ROHs to summarize long/short ROHs (see Supplementary Figs. 8–10). Our analysis suggests a different distribution of ROH lengths compared to our initial findings, revealing a mix of both short and long ROHs that aligns more closely with theoretical expectations given the population history we have described. We believe these changes have significantly strengthened our study and we are grateful for your constructive feedback.

This observation is supported by the large inferred inbreeding coefficient – if F is really 0.85 as calculated in Fig. 2d using observed and expected het, then there should be LOTS of long ROHs. Another

possible explanation for this excessively high F is, however, population structure, and I think the authors need to address whether this could be a problem for their analyses (see below).

Reply: Thanks for your critical comment. As mentioned above, we re-analyzed ROHs using more relaxed criteria to define ROH breakpoints. This approach aimed to address your concern that our initial criteria might have been too strict, potentially leading to the fragmentation of ROHs and an overestimation of short ROHs at the expense of recognizing fewer, longer ROHs. Overall, although we find a lower number of long rather than short ROHs, long ROHs constitute a large fraction of the Cat Ba langur genome (Supplementary Figs. 8–10). Regarding your observation of the large inferred inbreeding coefficient ($F=0.85$) and its implications for the presence of long ROHs, we have taken your feedback into serious consideration. Our inbreeding coefficient F is based on the whole-genome ROHs, including short and long ones. Population structure as a potential explanation for the high F is unlikely as gene flow among the three extant sub-populations still occurs as males are known to regularly move between sub-populations.

Perhaps this can also be affected by mapping to a distant reference, but I could not keep track of whether the ROH estimation was based on mapping to the close or to the distant reference genome. If the latter, it is almost certain that many spurious hets will be present to break up the ROHs into artificially short ROHs. One way to address the accuracy of inferred ROHs and their length is to simply plot them along the chromosomes and see if some putative longer ROHs are very visually being chopped up by the Plink ROH identification criteria.

Reply: Thanks for your comment. We added to the Results now a subsection “Sampling and datasets” in which we explain in detail which mapping data were used for which analysis. To avoid the effect of spurious hets to break up ROHs, we used for genome-wide ROH analysis the more closely Tfra_2.0 reference genome. Unfortunately, the Tfra_2.0 reference genome is not annotated to chromosome level, so that plotting ROHs along chromosomes is not possible.

I have a hard time understanding the authors’ claim that heterozygosity is 30-80 times higher in coding than in non-coding regions for the langurs. This is totally unexpected, regardless of any selection scenario that I can imagine. I tried to follow the authors calculation provided in Table S10, but this did not make any sense to me in relation to what is described in the methods. For example, I do not understand what the fixed numerator in the final two columns of Table S10 are. Also, I don’t understand why some individuals have very different numbers of heterozygous calls than their conspecific individuals. E.g. *T. poliocephalus* 1 has nearly 5 times as many heterozygous sites as *T. poliocephalus* 3. These results and calculations are not explained anywhere, so I found it hard to follow the authors’ calculation of heterozygosity, both in general and inside/outside coding regions.

Reply: Thank you for your attention to the heterozygosity patterns in our study and for your feedback. We re-calculated heterozygosity in protein-coding versus non-protein-coding regions by strictly using

exon versus non-exon data and obtained results that largely differ from our previous findings. Overall, we find in all limestone langurs a generally lower heterozygosity in coding versus non-coding regions. However, the Cat Ba langur shows in coding regions the highest heterozygosity rate among limestone langurs (Supplementary Fig. 6; Supplementary Table 7) and also the ratio of non-synonymous to synonymous variants was significantly increased (Supplementary Fig. 7; Supplementary Table 7), which is in line with predictions that in small populations the proportion of non-synonymous variants can increase due to weakened selection.

In relation to the above, it was unclear to me why the authors used ANGSD and realSFS for this analysis, as they have high-depth data and genotype calls from a GATK pipeline. As I understand the authors, they obtained a vcf file including genotype calls in nonvariant sites, which could have been used for the above analysis.

Reply: Thank you for your feedback regarding our choice of analysis tools. We re-ran the analysis using the GATK best practice pipeline and its associated VCF file, which includes genotype calls at nonvariant sites. The results obtained from the GATK pipeline align with our previous findings and further support the observations presented in our manuscript.

Demographic analyses:

I don't think it is prudent to use e.g. stairway plots (or other, purely SFS based analyses) to draw conclusions about demographic history on such a short scale as the authors do. I do not believe that SFSes can truly inform us about events within the last 100 years or even decades, as in Fig. 1e. At least I have never seen this before, and many papers highlight the poor performance of Stairway plots (and similar methods) in the very recent and very remote past, especially when sample sizes are so tiny. In the same vein, I think the authors need to show the underlying SFSes, because there could be issues related to e.g. population structure. The authors highlight that these samples are taken from different localities ("subpopulations"), and these demographic analyses are all super sensitive to population structure.

Reply: We appreciate your concerns regarding the use of SFS-based methods, as well as the potential impact of population structure on our results. We searched for publications investigating recent population histories and performed additional analyses with MSMC2, GONE, SMC++. We recognize the limitations of all these methods in the context of our dataset (only 4 genomes) and your valid concerns about their applicability on short timescales. After careful consideration and taking into account your and Reviewer's 1 comments, we decided to remove the entire demographic history part from the manuscript.

Genetic load:

The authors claim that the lower ratio of homozygous LoF/synonymous in Cat Ba langurs can be explained by purging (L248), but I don't think this is what would happen by purging – purging would perhaps (although this is not necessarily true) lead to a lower frequency of LoF variants relative to some other types of variants because recessive LoF are being removed by selection, but it should not lead to fewer HOMOZYGOUS LoF. Furthermore, the authors do not find a lower frequency of High impact deleterious alleles in Cat Ba langurs than other species (Fig. S15) , which is not consistent with accelerated purging in this population.

Reply: Thank you for your comments and for highlighting the potential misunderstanding regarding the impact of purging on the ratio of homozygous LoF to synonymous variants. Upon reviewing your feedback and the previous comments, we recognized a methodological oversight in our initial analysis. Consequently, we re-analyzed the part on deleterious mutations using mainly the Mmul_10 reference genome. We used genomic evolutionary rate profiling (GERP) scores and after filtering, we investigated the impact of deleterious mutation on protein function using four effect categories. In our results, we found that the Cat Ba langur had a significantly lower masked, but a significantly higher realized load than the other two limestone langur species (Supplementary Fig. 12). Also, the Cat Ba langur exhibited a significantly higher rate of homozygous high-impact deleterious mutations than the other two species (Fig. 3b; Supplementary Tables 9 and 10). The manuscript was revised accordingly.

In Table S12 I noticed that the depth of the homozygous LoF variants vary a lot even when looking across sites within the same individual. For example, DP for Tpol1 ranges between 5-128. This to me is a potential sign of problems, and I noticed that the authors only imposed a depth filter of minimum 4 when calling genotypes (L435). I would suggest choosing a narrower depth range for filtering SNPs, as very low or very high coverage regions potentially arise from mapping problems.

Reply: We appreciate your observation regarding the depth variation in homozygous LoF variants within the same individual and your valuable suggestion to impose a narrower depth range for SNP filtering. Consequently, we have revisited our methods and made the necessary adjustments to address this concern. Specifically, we have tightened the depth range (10-150) for filtering SNPs to exclude regions with very low or very high coverage, which may indeed arise from mapping problems. This refinement did not change our results much. However, we omitted the LoF part now completely as we performed more straightforward analysis to infer genetic load (see reply to comment before).

Positive selection and adaptation:

I would argue that the sample sizes are problematically low for some of the scan statistics that the authors apply. Furthermore, given the considerable evolutionary distance to the comparison population, some measures (certainly F_{st}) seem like an odd choice to infer positive selection. I am assuming that F_{st} genome-wide between the two langur species is close to 1. Outlier scans based on statistics with very narrow distributions will not work very well, and will be strongly affected by noise e.g. related to low

sample sizes. If the inbreeding coefficient is really as high as estimated by the authors, this will lead to many spurious windows of low genetic diversity, making this a poor scan statistic as well (in addition to the effect of small sample size). I wonder why the authors did not use standard measures from comparative genomics, such as dN/dS ratios, to infer positive selection in the Cat Ba langur, given the low sample sizes and long evolutionary distances to the comparison populations? The process of taking the overlap of outliers from different methods is pretty ad hoc, and it should be acknowledged that the false-positive rate of such analyses is unknown (top 5% is quite inclusive).

Reply: Thanks for your comment. Indeed, comparative genomic methods are more convincing and eliminate noise, but we have no high-quality genome of a Cat Ba langur generated by long-read sequencing. Here, although only four samples available, the methods we used for positive selection analysis (FST based on divergence, TajimaD based on neutral theory, Hp based on heterozygosity, XP-EHH and iHs based on haplotype) are quite strict to infer selective regions at in-group level and are useful to find differences between Cat Ba langur and other limestone langurs. We combined all methods and set the top 5% as threshold and picked the overlap as candidate genes under positive selection. We also analyzed the data using the top 1% as threshold, but no genes remained. Overall, we followed here standard methods (as, for example, described in this review: <https://www.frontiersin.org/journals/genetics/articles/10.3389/fgene.2014.00293/full>) and therefore consider our analyses as reliable.

Some of the analyses performed to identify unique adaptations are rather ad hoc or informal. For example, the finding of a single fixed LoF variant in gene *CDH26* is taken as putative evidence of positive selection. While this is possible, there are other evolutionary processes that could lead to the same result, e.g. inbreeding (which the authors have reported to be very high) or lack of sufficiently efficient negative selection. In fact, in the genetic load section of the study, the authors implicitly assume that LoF variants are under negative selection. The authors should discuss these limitations, and/or supplement them with more formal statistical tests for positive selection. In addition, the functional role of this mutation as an adaptation to saltwater intake is speculative and should be presented as such.

Reply: Thanks for your comment and we are sorry that you had the impression some of the analyses are rather ad hoc or informal. Yes, we identified *CDH26* as LoF gene, but we did not speculate if this is due to positive or negative selection, or because of other evolutionary processes. We fully agree that it remains open if the mutation in *CDH26* really contributed to saltwater tolerance. However, this gene is linked to different adaptation to salinities also in other vertebrates, so its function in salinity adaptation is indicated. We investigated further genes known to be involved in adaptation to different salinities in other vertebrates and in ten of them, we found at least one non-synonymous variant fixed in the Cat Ba langur (Supplementary Table 19). Considering these newly identified genes, *CDH26* and various genes linked to sodium metabolism derived from enrichment analyses, we believe that these genes in combination give sufficient indications for a potential adaptation of Cat Ba langurs to saltwater consumption, which we present as speculation and not evidence.

In a similar vein, the results presented as indicative of “Enhanced climbing ability” are extremely speculative. Either the authors should remove this section, temper their interpretation substantially or substantiate their conclusion by additional experiments or data.

Reply: Thank you for this comment. After careful consideration, we have decided to remove this part entirely from the manuscript.

Reviewers' Comments:

Reviewer #1:

Remarks to the Author:

The focus of the manuscript was to explore genetic diversity and genomic adaptation of a critically endangered langur species, the Cat Ba langur (*Trachypithecus poliocephalus*). To do so the authors carried out whole genome resequencing of four individuals. The authors combined previous sequenced individuals of two closely related langur species (*T. francoisi* and *T. leucocephalus*), and employed a series of population and conservation genomic analyses. The authors were able to obtain divergence times, population structure, gene flow among the three species. Moreover, they were able to estimate inbreeding level through analyses of ROH and genetic load in order to explore the genetic legacy of an endangered species. The most interesting part of the study is that they employed several analyses to detect candidate genes involved in Cat Ba langur's adaptive adaptation, such as high calcium intake, high salt water consumption and climbing movement. Overall, the authors define a picture of the evolution of an endangered primate species.

General Comments:

Overall, the article includes a considerable amount of analyses that is brought to bear on how a critically endangered primate evolve in an unique landscape. The article is a valuable contribution to our understanding of conservation genomics and adaptive evolution of extremely small population. However, I feel the two main themes of the study, i.e. genomic legacy of a small population and local adaptation is not well connected. Some of analyses (i.e. Population structure, PCA and D-statistics) are not necessary to support these two main themes. Justifications of doing these are missing. For the conservation genomic session, I wonder whether the authors are able to separate the overall genetic load to realized and masked load, separately, whether they can annotated the function of deleterious mutations in order to determine potential risk of the population. The functional analysis of detecting candidate focused on very specific life styles and behaviour of this species, which were already targeted by the authors in their previous studies. I'm wondering whether there are other selective regions involved the divergence of the species. The vast majority of my comments are attempts to improve clarity in parts of the manuscript. I would encourage the authors to rewrite portions of the Results and Discussion, especially the Results part, which are mixed with methods, results and interpretation of results, making it is a bit difficult to follow. Many of the ideas in the Results are valuable, but these ideas are not always described well and in many cases the ideas are not connected to each other well. A focused revision of the discussion would considerably improve the manuscript.

Specific Comments

Abstract:

Line 64: Is chromosome 19 unique in all primates, or just in langur species?

Introduction :

Line 89: Would be nice to show a photo of the study species in Figure 1 in order to make readers to see the species.

The introduction is very short. A nice summary of existing knowledge of the evolutionary history of the three langur species will be helpful. Whether their distributions are overlapped (at least historically), which may lead to the likelihood of historical introgression among them. I think these information is useful to formulate scientific questions.

L 106-114: Better to propose specific research questions here based on previous studies.

Results:

L 117-153: Because of lacking justifications and background information in introduction, I feel some analyses of these parts are not necessary. For instance, why PCA and admixture are useful?

L 154-173: To my opinion, the recent demographic history is more important for the study species. PSMC definitely has no power to illustrate this. I suggest the authors used more methods to approve it. For instance, SMC++ and GONE may be suitable methods. Further the divergence models among the three species can be inferred in more sophisticated approaches, such as fastsimcoal. However, the small sample size of the study might limit the use of this modelling approach.

L 167: Remove “individuals” as N_e has no unit.

L 175-188: The whole part is very descriptive. Authors may summarize these comparisons.

L 189-200: Would be nice to see the correlation between the length of ROH and number of ROH, because it can be also suggestive of historical bottleneck.

L 203: I do not find any discussion about the ‘magic chromosome 19’. Why it is special?

L 241: Would it be possible to know the function of genes containing deleterious mutations?

L 255: I feel the tests of exploring genes under selection are robust as authors used six different methods. As I said in the general comments, these results were targeted to calcium intake adaptation, saltwater tolerance, climbing adaptation. Whether other interesting genes were detected?

L 365: This part is simply the Conclusion of the study. Not Discussion per se.

Methods:

L 423-425: Both mapping protocols using different reference genomes were employed to admixture analyses? I’m a bit confused here.

L 547: Beside F_{st} and Tajima’s D , d_{xy} and local recombination rate might be other choice of measurement to detect signal of selection. Did authors consider to combine them into the analysis?

Reviewer #2:

Remarks to the Author:

This is a review of the manuscript: “Genomic adaptation to small population size and saltwater consumption in the critically 2 endangered Cat Ba langur” by Zhang et al. The study performs population genetic analyses of the endangered Cat Ba langur. It is highly relevant to perform a population genetic study on this species, given its precarious conservation status and unique biology. As such the study is timely, but I found several analytical concerns that need to be addressed for this manuscript to become ready for publication, in my opinion. Some of these lead to conclusions that are either unexpected or unreliable, given the limitations of the methods

applied.

ROH estimation, inbreeding and diversity:

I am skeptical that the authors have been able to accurately infer ROHs, particularly their length distributions. Given the authors finding of a continuously declining population size, I see no obvious explanation for why most ROHs should be short (100kb-1mb). I think the authors should reconsider whether they are using too strict criteria for breaking up ROHs, and therefore erroneously inferring many short rather than fewer long ROHs.

This observation is supported by the large inferred inbreeding coefficient – if F is really 0.85 as calculated in Fig. 2d using observed and expected het, then there should be LOTS of long ROHs. Another possible explanation for this excessively high F is, however, population structure, and I think the authors need to address whether this could be a problem for their analyses (see below). Perhaps this can also be affected by mapping to a distant reference, but I could not keep track of whether the ROH estimation was based on mapping to the close or to the distant reference genome. If the latter, it is almost certain that many spurious hets will be present to break up the ROHs into artificially short ROHs. One way to address the accuracy of inferred ROHs and their length is to simply plot them along the chromosomes and see if some putative longer ROHs are very visually being chopped up by the Plink ROH identification criteria.

I have a hard time understanding the authors' claim that heterozygosity is 30-80 times higher in coding than in non-coding regions for the langurs. This is totally unexpected, regardless of any selection scenario that I can imagine. I tried to follow the authors calculation provided in Table S10, but this did not make any sense to me in relation to what is described in the methods. For example, I do not understand what the fixed numerator in the final two columns of Table S10 are. Also, I don't understand why some individuals have very different numbers of heterozygous calls than their conspecific individuals. E.g. *T. poliocephalus* 1 has nearly 5 times as many heterozygous sites as *T. poliocephalus* 3. These results and calculations are not explained anywhere, so I found it hard to follow the authors' calculation of heterozygosity, both in general and inside/outside coding regions. In relation to the above, it was unclear to me why the authors used ANGSD and realSFS for this analysis, as they have high-depth data and genotype calls from a GATK pipeline. As I understand the authors, they obtained a vcf file including genotype calls in nonvariant sites, which could have been used for the above analysis.

Demographic analyses:

I don't think it is prudent to use e.g. stairway plots (or other, purely SFS based analyses) to draw conclusions about demographic history on such a short scale as the authors do. I do not believe that SFSes can truly inform us about events within the last 100 years or even decades, as in Fig. 1e. At least I have never seen this before, and many papers highlight the poor performance of Stairway plots (and similar methods) in the very recent and very remote past, especially when sample sizes are so tiny. In the same vein, I think the authors need to show the underlying SFSes, because there could be issues related to e.g. population structure. The authors highlight that these samples are taken from different localities ("subpopulations"), and these demographic analyses are all super sensitive to population structure.

Genetic load:

The authors claim that the lower ratio of homozygous LoF/synonymous in Cat Ba langurs can be explained by purging (L248), but I don't think this is what would happen by purging – purging would

perhaps (although this is not necessarily true) lead to a lower frequency of LoF variants relative to some other types of variants because recessive LoF are being removed by selection, but it should not lead to fewer HOMOZYGOUS LoF. Furthermore, the authors do not find a lower frequency of High impact deleterious alleles in Cat Ba langurs than other species (Fig. S15), which is not consistent with accelerated purging in this population.

In Table S12 I noticed that the depth of the homozygous LoF variants vary a lot even when looking across sites within the same individual. For example, DP for Tpol1 ranges between 5-128. This to me is a potential sign of problems, and I noticed that the authors only imposed a depth filter of minimum 4 when calling genotypes (L435). I would suggest choosing a narrower depth range for filtering SNPs, as very low or very high coverage regions potentially arise from mapping problems.

Positive selection and adaptation:

I would argue that the sample sizes are problematically low for some of the scan statistics that the authors apply. Furthermore, given the considerable evolutionary distance to the comparison population, some measures (certainly F_{st}) seem like an odd choice to infer positive selection. I am assuming that F_{st} genome-wide between the two langur species is close to 1. Outlier scans based on statistics with very narrow distributions will not work very well, and will be strongly affected by noise e.g. related to low sample sizes. If the inbreeding coefficient is really as high as estimated by the authors, this will lead to many spurious windows of low genetic diversity, making this a poor scan statistic as well (in addition to the effect of small sample size). I wonder why the authors did not use standard measures from comparative genomics, such as dN/dS ratios, to infer positive selection in the Cat Ba langur, given the low sample sizes and long evolutionary distances to the comparison populations? The process of taking the overlap of outliers from different methods is pretty ad hoc, and it should be acknowledged that the false-positive rate of such analyses is unknown (top 5% is quite inclusive).

Some of the analyses performed to identify unique adaptations are rather ad hoc or informal. For example, the finding of a single fixed LoF variant in gene CDH26 is taken as putative evidence of positive selection. While this is possible, there are other evolutionary processes that could lead to the same result, e.g. inbreeding (which the authors have reported to be very high) or lack of sufficiently efficient negative selection.

Reviewer #1:

Remarks to the Author:

The authors have addressed my previous comments satisfactorily (for example, by removing some results that loosely connect to the main themes of the study). I also note that they have strived to address concerns by reorganising the text, specifically genetic load, and testing positive selection. The manuscript is well-considered and revised.

My final main comment is still about the limited sample size for testing positive selection. Despite authors employing various analyses based on different principles to identify selective genes, certain methods, like haplotype-based methods, are found to be susceptible to small sample sizes, confounding the signal arising from selection (see Klassmann & Gautier (2022), PLoS One; Fagny et al. (2014); Molecular Biology and Evolution). I propose that additional revisions should be made to address this constraint and provide a rationale for potential bias. The other reviewer suggested using dN/dS ratios to infer the signal of selection. I believe it is worthwhile to try.

Reviewer #2:

Remarks to the Author:

The authors made substantial changes to the manuscript, which I think has improved it. I therefore recommend acceptance of the manuscript, with a few suggestions for improvements.

Minor comment 1: I still think that the authors' ability to infer long ROHs appears compromised, if the longest ROHs found are 1.5mbp. It's also not that interesting to distinguish between very finely resolved ROH length bins spanning each only 250kbp, as the authors do in Fig. S10. I also think it's weird that ROH counts in *T. poliocephalus* decrease with length bins up to 1500kbp, and then suddenly becomes higher, but maybe this is because there are many really long ROHs (much longer than 1500kbp) identified in *T. poliocephalus*? In that case, I would choose different bins, such as 100-500kbp, 500-1000kbp, 1000-2000kbp, 2000-5000kbp and >5000kbp, but this is just a suggestion. I do believe the authors' qualitative statement that there are more and longer ROHs in *T. poliocephalus* than in the other species, but I would like the authors to caution that inferred ROH lengths are notoriously difficult and sensitive to filtering, and it looks like the authors may still be finding too many short ROHs.

Minor comment 2: I appreciate the author's response to the selection analyses comments. However, these results remain opaque, as none of the results from the different scan statistics are shown anywhere. I would encourage the authors to show them in the supplement, e.g. as Manhattan plots or similar. I still believe that most of the scan statistics are meaningless with such tiny sample sizes and lots of inbreeding. E.g. I would assume that at least 5% of the windows have $F_{st} = 1$ between two different species, and what does top 5% even mean then? Furthermore, sample sizes for F_{st} really are too small, which is also true for all the other scan statistics. E.g. Tajima's D in essence is based on site frequency spectrum, which is not very informative with 4 samples. This limitation will make each of the scan statistics more noisy and prone to type I errors, so it would be nice to see them individually and also some kind of Venn diagram or upset plot to see how often they agree and disagree, respectively.

Minor comment 3: The authors state in their reply that: "We added to the Results now a subsection "Sampling and datasets" in which we explain in detail which mapping data were used for which analysis". If the authors are referring to the text in L137-141, this is by no means a comprehensive explanation of which data goes into what analysis. For example, I was finding myself still wondering which mapping was used for the genetic load analyses, because that is still not explained anywhere. I recommend the authors to add a table somewhere with an overview of which data went into which analyses (ideally including what filtering was done prior to each analysis).
on. In fact, in the genetic load section of the study, the authors implicitly assume that LoF variants are under negative selection. The authors should discuss these limitations, and/or supplement them with more formal statistical tests for positive selection. In addition, the functional role of this mutation as an adaptation to saltwater intake is speculative and should be presented as such. In a similar vein, the results presented as indicative of "Enhanced climbing ability" are extremely

speculative. Either the authors should remove this section, temper their interpretation substantially or substantiate their conclusion by additional experiments or data.

Comments by the Reviewers:**Reviewer #1:**

The authors have addressed my previous comments satisfactorily (for example, by removing some results that loosely connect to the main themes of the study). I also note that they have strived to address concerns by reorganising the text, specifically genetic load, and testing positive selection. The manuscript is well-considered and revised.

My final main comment is still about the limited sample size for testing positive selection. Despite authors employing various analyses based on different principles to identify selective genes, certain methods, like haplotype-based methods, are found to be susceptible to small sample sizes, confounding the signal arising from selection (see Klassmann & Gautier (2022), PLoS One; Fagny et al. (2014); Molecular Biology and Evolution). I propose that additional revisions should be made to address this constraint and provide a rationale for potential bias. The other reviewer suggested using Ka/Ks ratios to infer the signal of selection. I believe it is worthwhile to try.

Reply: Thank you very much for the positive feedback and the additional comments about positive

selection. In light of the comments by the editor and both reviewers concerning positive selection analysis, we decided to remove some of the positive selection methods (iHS, Tajima's D , H_p and F_{ST}) because of their weak performance with small sample size, but added instead Ka/Ks ratios. Our positive selection analysis is now based on three methods (Ka/Ks ratio, XP-EHH and $\theta\pi$), which, however, revealed no genes under positive selection related to calcium metabolism. However, we identified various fixed non-synonymous mutations in genes related to calcium and sodium metabolism, which are in our opinion strong indications for potential adaptation to increased calcium intake and saltwater consumption. Nevertheless, we toned down the respective parts in the Results, Discussion and Conclusion sections.

Reviewer #2:

The authors made substantial changes to the manuscript, which I think has improved it. I therefore recommend acceptance of the manuscript, with a few suggestions for improvements.

Minor comment 1: I still think that the authors' ability to infer long ROHs appears compromised, if the longest ROHs found are 1.5mbp. It's also not that interesting to distinguish between very finely resolved ROH length bins spanning each only 250kbp, as the authors do in Fig. S10. I also think it's weird that ROH counts in *T. poliocephalus* decrease with length bins up to 1500kbp, and then suddenly becomes higher, but maybe this is because there are many really long ROHs (much longer than 1500kbp) identified in *T. poliocephalus*? In that case, I would choose different bins, such as 100-500kbp, 500-1000kbp, 1000-2000kbp, 2000-5000kbp and >5000kbp, but this is just a suggestion. I do believe the authors' qualitative statement that there are more and longer ROHs in *T. poliocephalus* than in the other species, but I would like the authors to caution that inferred ROH lengths are notoriously difficult and sensitive to filtering, and it looks like the authors may still be finding too many short ROHs.

Reply: Thank you very much for the positive feedback and the additional helpful comments. We apologize for any confusion concerning our ROH analysis, but we found ROHs much longer than 1.5 Mb, which was probably not clear before. The longest ROHs in limestone langur individuals had lengths of 25.48-40.16 Mb (*I. poliocephalus*), 10.31-22.81 Mb (*I. leucocephalus*), 7.68-16.59 Mb (*I. francoisi*), 13.11 Mb (*I. laotum*), 11.12 Mb (*I. hatinhensis*), and 7.09 Mb (*I. ebenus*). This information is now provided in the Results and Discussion sections. In three of the rainforest langurs, we find even much longer ROHs (*I. obscurus*: 130.98 Mb, *I. auratus*: 130.98 Mb, *I. germaini*: 122.77 Mb; data not shown), which is likely the result of breeding between closely related individuals in captivity. As we find such long ROHs, we believe that our methods to infer ROHs are correct and reliable. In order to detect/describe long ROHs, we followed your suggestion to use other bins (presented now in Supplementary Figs. 9 and 10).

Minor comment 2: I appreciate the author's response to the selection analyses comments. However, these results remain opaque, as none of the results from the different scan statistics are shown anywhere. I would encourage the authors to show them in the supplement, e.g. as Manhattan plots or similar. I still believe that most of the scan statistics are meaningless with such tiny sample sizes and lots of inbreeding. E.g. I would assume that at least 5% of the windows have $F_{ST} = 1$ between two different species, and what does top 5% even mean then? Furthermore, sample sizes for F_{ST} really are

too small, which is also true for all the other scan statistics. E.g. Tajima's D in essence is based on site frequency spectrum, which is not very informative with 4 samples. This limitation will make each of the scan statistics more noisy and prone to type I errors, so it would be nice to see them individually and also some kind of Venn diagram or upset plot to see how often they agree and disagree, respectively.

Reply: Thank you for your critical and helpful comment. In light of the comments by the editor and both reviewers concerning positive selection analysis, we decided to remove some of the positive selection methods (iHS, Tajima's D, *Hp* and *F_{ST}*) because of their weak performance with small sample size, but added instead Ka/Ks ratios. Our positive selection analysis is now based on three methods (Ka/Ks ratio, XP-EHH and $\theta\pi$), which, however, revealed no genes under positive selection related to calcium metabolism. However, we identified various fixed non-synonymous mutations in genes related to calcium and sodium metabolism, which are in our opinion strong indications for potential adaptation to increased calcium intake and saltwater consumption. Nevertheless, we toned down the respective parts in the Results, Discussion and Conclusion sections. A Venn diagram with the number of positively selected genes revealed by the three applied methods and their overlaps is presented in Supplementary Fig. 13.

Minor comment 3: The authors state in their reply that: "We added to the Results now a subsection "Sampling and datasets" in which we explain in detail which mapping data were used for which analysis". If the authors are referring to the text in L137-141, this is by no means a comprehensive explanation of which data goes into what analysis. For example, I was finding myself still wondering which mapping was used for the genetic load analyses, because that is still not explained anywhere. I recommend the authors to add a table somewhere with an overview of which data went into which analyses (ideally including what filtering was done prior to each analysis).

Reply: Thank you for your valuable feedback. We sincerely apologize for any confusion caused by our previous explanation. To address your concern, we have now added an additional supplementary table (Supplementary Table 3) in which we show which mapping data were used for which analysis. We hope this addition provides more clarity.

Reviewers' Comments:

Reviewer #1:

Remarks to the Author:

The focus of the manuscript was to explore genetic diversity and genomic adaptation of a critically endangered langur species, the Cat Ba langur (*Trachypithecus poliocephalus*). To do so the authors carried out whole genome resequencing of four individuals. The authors combined previous sequenced individuals of two closely related langur species (*T. francoisi* and *T. leucocephalus*), and employed a series of population and conservation genomic analyses. The authors were able to obtain divergence times, population structure, gene flow among the three species. Moreover, they were able to estimate inbreeding level through analyses of ROH and genetic load in order to explore the genetic legacy of an endangered species. The most interesting part of the study is that they employed several analyses to detect candidate genes involved in Cat Ba langur's adaptive adaptation, such as high calcium intake, high salt water consumption and climbing movement. Overall, the authors define a picture of the evolution of an endangered primate species.

General Comments:

Overall, the article includes a considerable amount of analyses that is brought to bear on how a critically endangered primate evolve in an unique landscape. The article is a valuable contribution to our understanding of conservation genomics and adaptive evolution of extremely small population. However, I feel the two main themes of the study, i.e. genomic legacy of a small population and local adaptation is not well connected. Some of analyses (i.e. Population structure, PCA and D-statistics) are not necessary to support these two main themes. Justifications of doing these are missing. For the conservation genomic session, I wonder whether the authors are able to separate the overall genetic load to realized and masked load, separately, whether they can annotated the function of deleterious mutations in order to determine potential risk of the population. The functional analysis of detecting candidate focused on very specific life styles and behaviour of this species, which were already targeted by the authors in their previous studies. I'm wondering whether there are other selective regions involved the divergence of the species. The vast majority of my comments are attempts to improve clarity in parts of the manuscript. I would encourage the authors to rewrite portions of the Results and Discussion, especially the Results part, which are mixed with methods, results and interpretation of results, making it is a bit difficult to follow. Many of the ideas in the Results are valuable, but these ideas are not always described well and in many cases the ideas are not connected to each other well. A focused revision of the discussion would considerably improve the manuscript.

Specific Comments

Abstract:

Line 64: Is chromosome 19 unique in all primates, or just in langur species?

Introduction :

Line 89: Would be nice to show a photo of the study species in Figure 1 in order to make readers to see the species.

The introduction is very short. A nice summary of existing knowledge of the evolutionary history of the three langur species will be helpful. Whether their distributions are overlapped (at least historically), which may lead to the likelihood of historical introgression among them. I think these information is useful to formulate scientific questions.

L 106-114: Better to propose specific research questions here based on previous studies.

Results:

L 117-153: Because of lacking justifications and background information in introduction, I feel some analyses of these parts are not necessary. For instance, why PCA and admixture are useful?

L 154-173: To my opinion, the recent demographic history is more important for the study species. PSMC definitely has no power to illustrate this. I suggest the authors used more methods to approve it. For instance, SMC++ and GONE may be suitable methods. Further the divergence models among the three species can be inferred in more sophisticated approaches, such as fastsimcoal. However, the small sample size of the study might limit the use of this modelling approach.

L 167: Remove “individuals” as N_e has no unit.

L 175-188: The whole part is very descriptive. Authors may summarize these comparisons.

L 189-200: Would be nice to see the correlation between the length of ROH and number of ROH, because it can be also suggestive of historical bottleneck.

L 203: I do not find any discussion about the ‘magic chromosome 19’. Why it is special?

L 241: Would it be possible to know the function of genes containing deleterious mutations?

L 255: I feel the tests of exploring genes under selection are robust as authors used six different methods. As I said in the general comments, these results were targeted to calcium intake adaptation, saltwater tolerance, climbing adaptation. Whether other interesting genes were detected?

L 365: This part is simply the Conclusion of the study. Not Discussion per se.

Methods:

L 423-425: Both mapping protocols using different reference genomes were employed to admixture analyses? I’m a bit confused here.

L 547: Beside F_{st} and Tajima’s D , d_{xy} and local recombination rate might be other choice of measurement to detect signal of selection. Did authors consider to combine them into the analysis?

Reviewer #2:

Remarks to the Author:

This is a review of the manuscript: “Genomic adaptation to small population size and saltwater consumption in the critically 2 endangered Cat Ba langur” by Zhang et al. The study performs population genetic analyses of the endangered Cat Ba langur. It is highly relevant to perform a population genetic study on this species, given its precarious conservation status and unique biology. As such the study is timely, but I found several analytical concerns that need to be addressed for this manuscript to become ready for publication, in my opinion. Some of these lead to conclusions that are either unexpected or unreliable, given the limitations of the methods

applied.

ROH estimation, inbreeding and diversity:

I am skeptical that the authors have been able to accurately infer ROHs, particularly their length distributions. Given the authors finding of a continuously declining population size, I see no obvious explanation for why most ROHs should be short (100kb-1mb). I think the authors should reconsider whether they are using too strict criteria for breaking up ROHs, and therefore erroneously inferring many short rather than fewer long ROHs.

This observation is supported by the large inferred inbreeding coefficient – if F is really 0.85 as calculated in Fig. 2d using observed and expected het, then there should be LOTS of long ROHs. Another possible explanation for this excessively high F is, however, population structure, and I think the authors need to address whether this could be a problem for their analyses (see below). Perhaps this can also be affected by mapping to a distant reference, but I could not keep track of whether the ROH estimation was based on mapping to the close or to the distant reference genome. If the latter, it is almost certain that many spurious hets will be present to break up the ROHs into artificially short ROHs. One way to address the accuracy of inferred ROHs and their length is to simply plot them along the chromosomes and see if some putative longer ROHs are very visually being chopped up by the Plink ROH identification criteria.

I have a hard time understanding the authors' claim that heterozygosity is 30-80 times higher in coding than in non-coding regions for the langurs. This is totally unexpected, regardless of any selection scenario that I can imagine. I tried to follow the authors calculation provided in Table S10, but this did not make any sense to me in relation to what is described in the methods. For example, I do not understand what the fixed numerator in the final two columns of Table S10 are. Also, I don't understand why some individuals have very different numbers of heterozygous calls than their conspecific individuals. E.g. *T. poliocephalus* 1 has nearly 5 times as many heterozygous sites as *T. poliocephalus* 3. These results and calculations are not explained anywhere, so I found it hard to follow the authors' calculation of heterozygosity, both in general and inside/outside coding regions. In relation to the above, it was unclear to me why the authors used ANGSD and realSFS for this analysis, as they have high-depth data and genotype calls from a GATK pipeline. As I understand the authors, they obtained a vcf file including genotype calls in nonvariant sites, which could have been used for the above analysis.

Demographic analyses:

I don't think it is prudent to use e.g. stairway plots (or other, purely SFS based analyses) to draw conclusions about demographic history on such a short scale as the authors do. I do not believe that SFSes can truly inform us about events within the last 100 years or even decades, as in Fig. 1e. At least I have never seen this before, and many papers highlight the poor performance of Stairway plots (and similar methods) in the very recent and very remote past, especially when sample sizes are so tiny. In the same vein, I think the authors need to show the underlying SFSes, because there could be issues related to e.g. population structure. The authors highlight that these samples are taken from different localities ("subpopulations"), and these demographic analyses are all super sensitive to population structure.

Genetic load:

The authors claim that the lower ratio of homozygous LoF/synonymous in Cat Ba langurs can be explained by purging (L248), but I don't think this is what would happen by purging – purging would

perhaps (although this is not necessarily true) lead to a lower frequency of LoF variants relative to some other types of variants because recessive LoF are being removed by selection, but it should not lead to fewer HOMOZYGOUS LoF. Furthermore, the authors do not find a lower frequency of High impact deleterious alleles in Cat Ba langurs than other species (Fig. S15), which is not consistent with accelerated purging in this population.

In Table S12 I noticed that the depth of the homozygous LoF variants vary a lot even when looking across sites within the same individual. For example, DP for Tpol1 ranges between 5-128. This to me is a potential sign of problems, and I noticed that the authors only imposed a depth filter of minimum 4 when calling genotypes (L435). I would suggest choosing a narrower depth range for filtering SNPs, as very low or very high coverage regions potentially arise from mapping problems.

Positive selection and adaptation:

I would argue that the sample sizes are problematically low for some of the scan statistics that the authors apply. Furthermore, given the considerable evolutionary distance to the comparison population, some measures (certainly F_{st}) seem like an odd choice to infer positive selection. I am assuming that F_{st} genome-wide between the two langur species is close to 1. Outlier scans based on statistics with very narrow distributions will not work very well, and will be strongly affected by noise e.g. related to low sample sizes. If the inbreeding coefficient is really as high as estimated by the authors, this will lead to many spurious windows of low genetic diversity, making this a poor scan statistic as well (in addition to the effect of small sample size). I wonder why the authors did not use standard measures from comparative genomics, such as dN/dS ratios, to infer positive selection in the Cat Ba langur, given the low sample sizes and long evolutionary distances to the comparison populations? The process of taking the overlap of outliers from different methods is pretty ad hoc, and it should be acknowledged that the false-positive rate of such analyses is unknown (top 5% is quite inclusive).

Some of the analyses performed to identify unique adaptations are rather ad hoc or informal. For example, the finding of a single fixed LoF variant in gene CDH26 is taken as putative evidence of positive selection. While this is possible, there are other evolutionary processes that could lead to the same result, e.g. inbreeding (which the authors have reported to be very high) or lack of sufficiently efficient negative selection.

Reviewer #1:

Remarks to the Author:

The authors have addressed my previous comments satisfactorily (for example, by removing some results that loosely connect to the main themes of the study). I also note that they have strived to address concerns by reorganising the text, specifically genetic load, and testing positive selection. The manuscript is well-considered and revised.

My final main comment is still about the limited sample size for testing positive selection. Despite authors employing various analyses based on different principles to identify selective genes, certain methods, like haplotype-based methods, are found to be susceptible to small sample sizes, confounding the signal arising from selection (see Klassmann & Gautier (2022), PLoS One; Fagny et al. (2014); Molecular Biology and Evolution). I propose that additional revisions should be made to address this constraint and provide a rationale for potential bias. The other reviewer suggested using dN/dS ratios to infer the signal of selection. I believe it is worthwhile to try.

Reviewer #2:

Remarks to the Author:

The authors made substantial changes to the manuscript, which I think has improved it. I therefore recommend acceptance of the manuscript, with a few suggestions for improvements.

Minor comment 1: I still think that the authors' ability to infer long ROHs appears compromised, if the longest ROHs found are 1.5mbp. It's also not that interesting to distinguish between very finely resolved ROH length bins spanning each only 250kbp, as the authors do in Fig. S10. I also think it's weird that ROH counts in *T. poliocephalus* decrease with length bins up to 1500kbp, and then suddenly becomes higher, but maybe this is because there are many really long ROHs (much longer than 1500kbp) identified in *T. poliocephalus*? In that case, I would choose different bins, such as 100-500kbp, 500-1000kbp, 1000-2000kbp, 2000-5000kbp and >5000kbp, but this is just a suggestion. I do believe the authors' qualitative statement that there are more and longer ROHs in *T. poliocephalus* than in the other species, but I would like the authors to caution that inferred ROH lengths are notoriously difficult and sensitive to filtering, and it looks like the authors may still be finding too many short ROHs.

Minor comment 2: I appreciate the author's response to the selection analyses comments. However, these results remain opaque, as none of the results from the different scan statistics are shown anywhere. I would encourage the authors to show them in the supplement, e.g. as Manhattan plots or similar. I still believe that most of the scan statistics are meaningless with such tiny sample sizes and lots of inbreeding. E.g. I would assume that at least 5% of the windows have $F_{st} = 1$ between two different species, and what does top 5% even mean then? Furthermore, sample sizes for F_{st} really are too small, which is also true for all the other scan statistics. E.g. Tajima's D in essence is based on site frequency spectrum, which is not very informative with 4 samples. This limitation will make each of the scan statistics more noisy and prone to type I errors, so it would be nice to see them individually and also some kind of Venn diagram or upset plot to see how often they agree and disagree, respectively.

Minor comment 3: The authors state in their reply that: "We added to the Results now a subsection "Sampling and datasets" in which we explain in detail which mapping data were used for which analysis". If the authors are referring to the text in L137-141, this is by no means a comprehensive explanation of which data goes into what analysis. For example, I was finding myself still wondering which mapping was used for the genetic load analyses, because that is still not explained anywhere. I recommend the authors to add a table somewhere with an overview of which data went into which analyses (ideally including what filtering was done prior to each analysis).

Reviewers' Comments:

Reviewer #1

(Remarks to the Author)

First of all, I appreciate the authors' efforts in improving the dataset and results of the manuscript. Some of my major concerns have been solved. Overall, I think the quality of the manuscript is greatly improved and suitable for consideration by Nature Communications. My only concern is about the phylogeny of *Trachypithecus* you derived. The phylogeny is incomplete as it does not include all 16 species of the genus. Kindly submit information regarding the number of species within the genus and the number of species within the limestone langur group. Lines 94-95 would be an appropriate location to incorporate this information. To entertain our readers, including species cartoons or illustrations beside the species labels in Figure 1c may be beneficial.

(Remarks to the Editor)

Reviewer #2

(Remarks to the Author)

I think the authors have made a large effort to take action on the reviewer comments through several rounds of review, and I have no further comments. I therefore recommend acceptance of the manuscript. In fact, in the genetic load section of the study, the authors implicitly assume that LoF variants are under negative selection. The authors should discuss these limitations, and/or supplement them with more formal statistical tests for positive selection. In addition, the functional role of this mutation as an adaptation to saltwater intake is speculative and should be presented as such.

In a similar vein, the results presented as indicative of "Enhanced climbing ability" are extremely speculative. Either the authors should remove this section, temper their interpretation substantially or substantiate their conclusion by additional experiments or data.

Reviewer #1 (Remarks to the Author):

First of all, I appreciate the authors' efforts in improving the dataset and results of the manuscript. Some of my major concerns have been solved. Overall, I think the quality of the manuscript is greatly improved and suitable for consideration by Nature Communications. My only concern is about the phylogeny of *Trachypithecus* you derived. The phylogeny is incomplete as it does not include all 16 species of the genus. Kindly submit information regarding the number of species within the genus and the number of species within the limestone langur group. Lines 94-95 would be an appropriate location to incorporate this information. To entertain our readers, including species cartoons or illustrations beside the species labels in Figure 1c may be beneficial.

Reply: Many thanks for your comments. Following your suggestion, we added information about the number of species within *Trachypithecus* (22 not 16) and specifically within the limestone langur group (Introduction, paragraph 4). "*Trachypithecus poliocephalus* is a species of the colobine genus *Trachypithecus* which contains a total of 22 species, grouped into four species groups³¹⁻³⁴. *Trachypithecus poliocephalus* is one of the seven species of the *T. francoisi* or limestone langur group³¹⁻³⁴." Moreover, we added drawings of the investigated langur species in Figure 1c.

Reviewer #2 (Remarks to the Author):

I think the authors have made a large effort to take action on the reviewer comments through several rounds of review, and I have no further comments. I therefore recommend acceptance of the manuscript.

Reply: We highly appreciate your comments both in the past and present.